# Integrating Fireline Observations to Characterize Fire Plumes During Pyroconvective Extreme Wildfire Events: Implications for Firefighter Safety and Plume Modeling

Marc Castellnou Ribau[1,3], Mercedes Bachfischer[1], Pau Guarque[1], Laia Estivill[1], Marta Miralles Bover[1], Borja Ruiz[1], Jordi Pagès[3], Brian Verhoeven[2], Zisoula Ntasiou[4], Ove Stokkeland [5], Chiel van Heerwaarden[3], Tristan Roelofs[3], Martin Janssens[3], Cathelijne R. Stoof [3] and Jordi Vilà-Guerau de Arellano[3]

[1] GRAF. Catalan Fire and Rescue Service. Av Serragalliners, 08193 Cerdanyola del Vallés, Catalonia. Spain.
[2] Netherlands Institute for Public Safety. IJsselburcht 2, 6825 BS Arnhem, The Netherlands.
[3] Wageningen University & Research. PO box 47, 6700 AA Wageningen, The Netherlands.
[4] Hellenic Fire and Rescue Service. Mourouzi 4, 10674 Athens. Greece.
[5] Grenland Fire and Rescue IKS. Hydrovegen 53, 3936 Porsgrunn, Vestfold og Telemark, Norway.

*Correspondence to*: Marc Castellnou (mcastellnou@gencat.cat)

**Abstract.** Firefighter entrapments occur when wildfires suddenly transition into extreme wildfire events (EWEs). These transitions are often caused by pyroconvective fire-atmosphere coupling, triggered by a combination of high fire intensity and atmospheric vertical thermodynamic structure. Pyroconvection indices calculated using coarse atmospheric modeling data crudely detect these dynamic transitions due to highly localized atmospheric processes and changes in atmospheric conditions caused by the fire. Consequently, fire managers may remain unaware that fire behavior intensification due to fire-atmosphere coupling is outdating the safety protocols in place. This study presents a new in-plume profiling methodology to improve the assessment of fire-atmosphere interaction dynamics in real-time. As proof of concept, we analyzed 173 successful sondes (148 in-plume) launched during the 2021-2025 fire seasons in Spain, Chile, Greece, and the Netherlands. As a strategy to measure the fire-atmosphere coupling, we propose simultaneously launching two radiosondes: one to measure ambient conditions and another to capture data within the plume updraft. Comparing these profiles, we measure in-situ and in-real time the modification of state variables by the fire-atmosphere interaction. These new observations and methodology improve our assessment of pyroconvection dynamics, demonstrating practical implications that support their use by incident management teams. It has the potential to enhance awareness of possible near-accidents and tactical failures during extreme pyroconvective wildfire events. Additionally, it offers a comprehensive observational dataset to improve pyroconvection nowcasting and advance research on fire-atmosphere interaction.

## 1 Introduction

Pyroconvection is a key driver in the escalation from wildfires to extreme wildfire events. While dry convection plumes effectively accelerate fire spread , it is the development of moist pyroconvection plumes by the formation of pyrocumulus and pyrocumulonimbus (pyroCu/Cb, AMS, 2023) that dramatically intensifies fire behavior. Deep pyroCu/Cb events amplify dry pyroconvective plume dynamics through powerful indrafts and downdrafts, triggering chaotic surges in spread rate, increasing massive and long-range spotting (embers ignite new fires at a distance) on the head and flanks, and generating deep flames and vortices (McRae et al., 2015; Peterson et al., 2017). These rapid, unpredictable changes can catch responders off guard, leaving them with little time to react. This can undermine suppression tactics and create significant risks for both responders and civilians. Tragically, the history of deadly entrapments under these conditions illustrates the severity of the problem (Cardil and Molina, 2015; Cruz et al., 2012; Lahaye et al., 2018; Page et al., 2019).

The conditions favoring such destructive wildfires are increasing due to climate change and human policies in landscape and fire management. Firefighters must prepare to better detect pyroconvection transitions.

Safety on the fireline hinges on effectively predicting fire spread, particularly by understanding conditions that have previously led to entrapments following sudden changes in fire behavior (Wilson, 1977). Insights from these experiences have shaped

protocols and orders to enhance crew awareness and prevent future incidents (Ziegler, 2007). The LACES protocol condenses critical lessons into the memorable acronym: Lookout, Awareness, Communications, Escape Route, and Safety Zone (Gleason, 1991). In this framework, lookout observations and awareness of pyroconvection conditions, using indices and models, play a vital role. However, transitions in pyroconvection, especially those involving pyroCu/Cb clouds, are affected by highly localized surface and free tropospheric processes, which are hard to predict (Peterson et al., 2017). This complexity makes

real-time monitoring of fire plumes and their environment from the fireline a difficult, yet essential safety measure to prevent accidents and fatalities.

Since the 1950s, fire managers have conducted ambient radiosonde profiling to assess the in-situ pyroconvection potential (McCutchan, 1982) during big wildfire events. Using the profiles, the Haines index (Haines, 1989) has become vital for informing firefighters about pyroconvective extreme fire risks, despite its limitations and reduced sensitivity (Potter, 2018).

The analysis of fire-atmosphere coupling has progressed to evaluating temperature as a function of pressure on skew-T diagrams to gauge pyroconvection potential (Goens & Andrews, 1998). This method is based on the observation that wildfires producing pyroCu/Cb clouds often occur in a well-mixed convective boundary layer and moist mid-troposphere, forming the basis for pyroconvection analysis using the parcel method (Jenkins, 2004; Lareau and Clements, 2016; Tory et al., 2018).

The advent of regional and global atmospheric models has transformed this practice, enabling predictions of pyrocloud

occurrence through various indices, including convective available potential energy adapted to wildfires (fireCAPE) (Potter and Anaya, 2015), the maximum integrated buoyancy (Leach and Gibson, 2021), and the pyroCu firepower threshold, PFT (Tory and Kepert, 2021).

Nevertheless, the coupling between fire and a turbulent atmosphere is much more complex than can be captured by single indices of the ambient environment. The increase in observations has led to higher-fidelity analyses of turbulent fire plumes

(Freitas et al., 2007; Paugam et al., 2016; Rio et al., 2010) and complex fire-atmosphere coupling models such as MESO-NH/Forefire or WRF-Sfire (Couto et al., 2024; Filippi et al., 2013; Kochanski et al., 2019). Those models are deepening our understanding of deep pyroconvection and its underlying physics.

A crucial aspect for firefighters is the enhanced understanding that modeling provides regarding the interaction between turbulent plumes and fire spread (Heilman, 2023). This understanding is influenced by factors such as the size of the flaming

zone (Badlan et al., 2021), and the dynamics involved in moist pyroconvection (pyroCb) models (Peterson et al., 2017).

Despite these advancements in modeling, practical applications for decision-making remain limited. This limitation stems from the constantly evolving dynamic relationship between fire and the atmospheric boundary layer (ABL), necessitating accurate data collection for effective fire management (Lareau et al., 2024; Prichard et al., 2019). Safety concerns related to operating near extreme fire fronts mean data collection primarily occurs through experimental fire campaigns like FireFlux and RxCadre

(Benik et al., 2023; Clements et al., 2015, 2019) involving low to moderate-intensity fires. These campaigns miss the complexities of fast-transitioning pyroconvective events. More recent campaigns shifted focus towards wildfires to collect more extreme fire behavior (Clements et al., 2018; Rodriguez et al., 2020). Innovative measurement methods, including UAVs (Brewer and Clements, 2020; Koch et al., 2018) and radar (Lareau et al., 2022; McCarthy et al., 2019) enhanced data collection during ongoing extreme fires. Nevertheless, challenges such as mobility, safety, funding, and data processing continue to

hinder progress during active fires.

We aim to develop a fireline data-gathering methodology using in-plume radiosondes with two main objectives: (a) to advance the understanding and representation of pyroconvection and its impact on extreme fire behavior, and (b) to provide fire managers with a real-time tool for assessing the likelihood of occurrence of different pyroconvection prototypes (Castellnou

et al., 2022).

Despite observing state variables profiles by means of sondes has been used for decades, their use in wildfire updrafts for real-time comparisons with ambient profiles is challenging. We need to assess whether the uncontrolled ascent trajectory of a sonde

can capture plume height and state variables across different fire intensities and help evaluate pyroconvection characteristics. By obtaining accurate vertical profiles of ambient and in-plume updraft conditions during the early stages of fire growth, we seek to capture the potential for plume-driven modifications in the state variables, raising awareness of pyroconvection conditions.

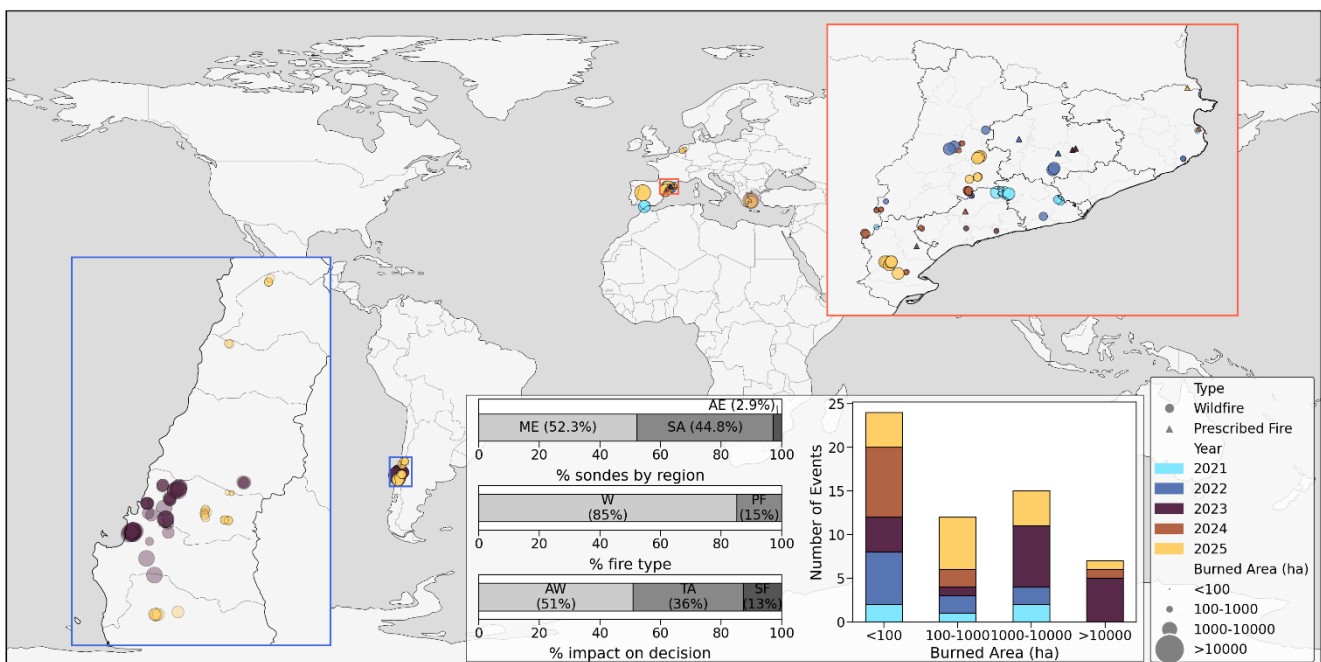

**Figure 1**: Characterisation of the 173 profile observations during the radiosonde campaigns conducted between 2021 and 2025. **(a)** Location of sondes, classified by their launch during wildfires (circles) or prescribed fires (triangles). The color of each dot represents the campaign year, while the size of the dot reflects the fire size (in hectares). **(b)** Regional distribution categorized as ME (Mediterranean Europe), AE (Atlantic Europe), and SA (South America). **(c)** Type of fire: wildfires (W) from prescribed fires (PF) **(d)** Sonde information impact on fire management classified as: Awareness for those assessing pyroconvection (AW), Tactical when the profiling information triggered adjustment of ongoing tactics (TA), and Entrapment for those that identified critical situations and led ultimately to safety evacuations (SF). **(e)** Summary of fire size by campaign year. Last updated on September 15, 2025 (Table S1).

## 2 Methodology

To develop a methodology for assessing pyroconvection during wildfire operations and to create a valuable dataset for improving models and research, we conducted field campaigns from 2021 to 2025 in Spain, Chile, Greece, and the Netherlands, launching 173 successful sondes (148 in-plume) during active fires (Figure 1, Table S1).

This approach was crafted through collaboration among firefighters, fire scientists, and meteorologists, prioritizing team safety and consistent data collection.

We outline the methodology, highlighting safety protocols, coordination, equipment selection, launching procedures, and data collection for vertical profiles in ambient conditions and plume updrafts.

### 2.1 Field campaigns

We focused on assisting fire managers in managing potential pyroconvection transitions to pyroCu/Cb. The vertical profile information gathered (Figure 1) was used to build awareness (51%), adjust tactics (36%), and avoid potential entrapments (13%).

To achieve this, we launched sondes across a wide range of fire sizes (Figure 1c) during its early stages of development, when pyroconvection was just being initiated, and the plume was still a surface or convective plume. We tested our methodology on both low-intensity prescribed fires (14.7%) and active wildfires (85.53%), including all vegetation types, including grasslands (16.2%), brushlands (40.5%), and forests (43.2%). Specifically, we targeted wildfires that have the potential to transition to pyroconvection during peak fire seasons: July to September in Mediterranean Europe (ME, 44.73%), March to May in Atlantic Europe (AE, 3.29%), and January to March in South America (SA, 51.98%).

## 2.2 Safety and coordination

Moving within the fire area requires adherence to safety protocols and coordination with the incident management team. We recommend deploying a sonde crew consisting of at least two members: a lookout and a launcher. This team will gather data and implement a LACES protocol with an emphasis on awareness.

Clear communication between the launch team and the aerial resources coordination is crucial to ensure safety during fire suppression operations involving helicopters and air tankers. The small colored sondes are safe for aircraft if their launch timing and position are known, as they mainly travel within the updraft of the plume, where aerial resources don't operate.

Before the launch, the team must select the Escape route and the Safety zone based on the expected fire behavior (Butler, 2014). These locations must be shared with the nearby firefighters, as they will be utilized for rescue efforts if necessary.

## 2.3 Equipment

Capturing information on the ongoing fire-atmosphere coupling to assess firefighter safety requires equipment capable of real-time, in-situ assessment of pyroconvection. To select the most suitable method, we compared the characteristics of five meteorological measurement techniques, namely professional high-altitude weather balloons, small weather balloons, doppler radar, unoccupied aerial vehicles (UAV), and helicopter sensors. Our requirements are as follows:

- Light, mobile equipment suitable to operate near the flame front and entirely operated by one person; a second person is only required for safe mobility and fire monitoring (lookout).
- Fast deployment within 5 minutes.
- In-situ and real-time information acquisition on the fireline, ready for immediate decision-making.
- Ensure compliance with specific safety requirements that may differ from general aerial control regulations. These are proposed by the fire service aerial coordination for operating alongside firefighting aerial resources: radiosondes weighing less than 50 grams and colored balloons with a capacity of less than 90 liters. Note that these requirements may vary internationally, and we adhere to the strictest standards
- Provision of two vertical profiles, one outside the fire's range of influence on the atmosphere, and one inside the fire plume to obtain the fire-modified vertical profile.
- Simultaneous, or ensemble measurements of atmospheric vertical profile thermodynamics up to lifting condensation level (LCL).
- Low cost. Affordable for the budget of firefighter crews.
- Low complexity: Implementing the methodology should be accessible and not require complex technical skills and knowledge

**Table 1**: Requirements for safe deployment in active wildfires and for providing real-time information on thermodynamic atmospheric profile conditions. Small balloons are the only equipment that meets all the specified requirements.

|  | Operational radiosonde systems | Small balloons | Radar Doppler | UAV-drones | Helicopter sensors |
|---|---|---|---|---|---|
| Max 2-people needed | X | X |  | X |  |
| < 5 min deployment | X | X |  | X | X |
| Real-time info | X | X |  |  |  |
| Aerial controller safety requirements |  | X | X |  | X |
| In fire/out fire profiles | X | X | X | X | X |
| Simultaneous measurement | X | X | X | X |  |

| | | | | | |
|---|---|---|---|---|---|
| Low cost | | X | | | |
| Ease of use | X | X | | | |

Comparison of professional high-altitude balloons, small balloons, radar doppler, UAV-drones, and helicopter sensors (Table 1) indicate that most tools were unreliable for rapid deployment in the fireline and provide real-time data with safety. The only exception are small balloons, which meet all the requirements listed above, and are safe enough for aerial resources in the unlikely case that the sonde travels near an aircraft.

We therefore selected a small helium balloon (60 liters), namely the light radiosondes model S1H2 (12 gr) from Windsond (Figure S1) to develop a measurement kit. The instrumental capabilities of the system have been previously tested against larger radiosonde systems, such as the RS41, during the LIAISE campaign (Boone, 2019) of ABL measurements in Lleida (Spain). Results showed a strong profile adjustment between both radiosonde systems (Castellnou et al., 2022). While certain weaknesses, such as a 40-meter altitude underestimation, issues with GPS processing, slow humidity response at cloud tops, and noisy wind profiles in turbulent conditions (Bessardon et al., 2019) were noted, they were not detrimental to the accuracy of identifying pyroconvective prototypes during wildfires (Castellnou et al., 2022).

To continuously validate the Windsond operational effectiveness, we systematically record plume measurements using fire service planes and radars whenever possible.

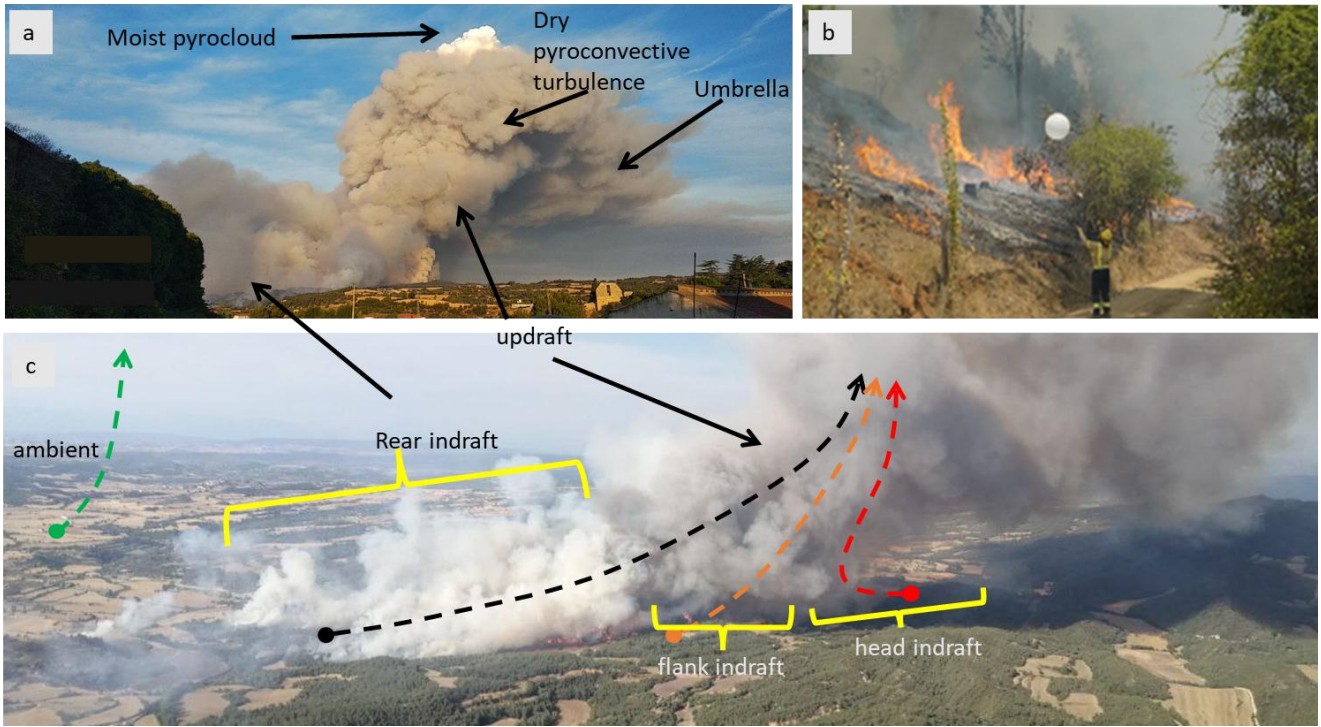

**Figure 2:** Sonde launching within the fire plume area. a) plume description from an upwind location. b) detail of a sonde launching (video S2.1 and S2.2). c) Launching locations relative to the indraft induced by the plume updraft. The black dashed arrow represents a rear indraft sonde, the orange dashed arrow represents a flank indraft sonde, the red dashed arrow represents a head indraft sonde, and the green dashed arrow represents the ambient sonde away from the plume's influence. This panel illustrates the key characteristics of a plume: the updraft, which conforms the chimney; the dry pyroconvection turbulence, inside the ABL and forming the grey smoke turbulence at the chimney's peak; the moist pyrocloud, defined by LCL and identified by where condensation occurs, typically forming pyroCu/Cb; and the umbrella, a dense layer of smoke that forms around the top of the updraft and extends downwind into the injection layer.

## 2.4 Strategy of the launching procedure and data workflow

Our strategy and primary objective were to systematically obtain (1) an ambient sonde outside the shading of the plume and (2) an in-plume or updraft sonde, launched to ascend inside the plume updraft cores, capturing the fire-induced changes in the ABL. Soundings should be taken no more than 1 hour apart (Figure 2) due to the ABL's response time of approximately one

hour or less (Granados-Muñoz et al., 2012; Liu & Liang, 2010; Stull, 1988). This maximum time ensures that the ambient and in-plume soundings remain comparable.

- Framing the day vertical atmospheric profile conditions with atmospheric numerical models:

After assessing the numerical model uncertainties of the GFS, ICON, and AROME models (Figure S2), we have chosen the ICON model, with a global horizontal resolution of 13×13 km, as our reference. With the European fires, we will

transition to the ICON-EU model, which offers a higher resolution of 7×7 km. The modeled atmospheric vertical profile provides a framework for the general conditions we can expect. (https://www.dwd.de/EN/ourservices/nwp_forecast_data/nwp_forecast_data.html).

- Criteria for maximum height sonde ascent:

We aim to reach altitudes defining the ABL and LCL before terminating the sonde for recovery. Given the elevated plume-

modified LCL height (Lareau and Clements, 2016), the balloon cut-off height is set at a minimum of 1000 m above the theoretical LCL, as indicated by atmospheric model data.

- Balloon filling-up:

We use a helium-pressurized container and a manometer installed on the fire service vehicle, systematically using 60 liters of helium to ensure the balloons have consistent characteristics.

- In-plume or updraft sonde:

Launched near the flame front into the plume's indraft, the sonde is carried by it into the plume base and ascends in the updraft cores. It measures state variables within the plume, affected by turbulent interactions between the fire and the atmosphere. Indraft intensity varies significantly from the head to the rear and flanks of the fire, influencing the transport into updraft cores and ultimately the sonde readings. To analyze the sensitivity of different indraft types to capturing the

characteristics of plume pyroconvection, we classified each in-plume sonde by position (Figure 2): head indraft (downwind of the fire front), flank indraft (on the flanks), and rear indraft (upwind). This classification ensures interoperability among sondes in the same indraft.

- Ambient sonde:

Launched outside the fire influence (Figure 2), it measures the vertical profile of the state variables in an environment

uninfluenced by the fire plume. By comparing data from both the in-plume descent and the ambient sondes, we can improve the reliability of our findings.

Although launching a separate ambient sonde is recommended, our campaign findings suggest that it may sometimes be operationally impractical. However, an ambient profile can also be obtained from the in-plume sonde descent path if the sonde is cut-down once it is outside the plume's influence. Although less reliable, analysis of such profiles measurements

taken during descent still enables us to identify key metrics in the fire-weather interaction, with acceptable variable uncertainty of less than 1 K in potential temperature and 2.2% in relative humidity (Figure S3).

The sonde operational workflow includes having the fire analyst as part of the launch team, enabling immediate analysis of observational data collected during the sounding. If the analyst is not present, data is uploaded from field mobile devices to

cloud storage for command-post analysis. The analyst reviews the vertical profiles to approve or adjust ongoing operations in collaboration with the incident commander and safety officer. Additional information is gathered from fireline crews, drones, planes, and meteorological radars, when available. Data management should occur within one hour of the in-plume launch. The process involves data transfer, profile visualization software, and a cloud archive to make the observations accessible to the incident management team.

## 2.5 Ambient, plume updraft, and fire spread data

### 2.5.1 Data collection for real-time monitoring of fire-atmosphere interaction

- In-situ radiosondes data (ambient and in-plume): The vertical profile variables (Table 2) of temperature $T^a$ (K), relative humidity RH (%), horizontal wind U (m·s$^{-1}$), and sonde rising velocity (m·s$^{-1}$) are retrieved at a 1-second resolution. Here, we use the sonde rising velocity as a proxy for vertical wind speed (w). The data is transformed to state or conserved variables: specific humidity q (gr·kg$^{-1}$), potential temperature θ (K), and virtual potential temperature θ$_v$ (K) ( S6).
- Instantaneous Fire Spread.
    - Observed rate of spread (ROS, m·s$^{-1}$).
    - Size of the head flaming zone and deep flame (m$^2$).

### 2.5.2 Data collection for post-analysis and research

- Radar measured echotop. It is a proxy measure for the plume top. We analyze radar echotop heights (m) using data from the Servei Català de Meteorologia (www.meteo.cat). We filter the radar echotop data and define the estimated plume top as the maximum height at which the reflectivity equals or exceeds 12 dBZ (Krishna et al., 2023). Unfortunately, the data for all fires is not available. This dataset is utilized to validate the estimates of plume tops collected from in-plume radiosondes during 18 wildfires
- Overall Fire Spread and intensity.
    - Fuel type (Scott & Burgan, 2005): We record the dominant fuel type to be used in heat flux modeling.
    - Fire isochrones. Produced by the Fire Service, it allows us to compute the rate of spread (ROS, m·s$^{-1}$) as the maximum distance in the wind direction between two consecutive hourly isochrones (Duane et al., 2024).
    - Fire Intensity: Using ROS and knowing the fuel type we estimate the heat flux (kW·m$^{-2}$) and the fireline intensity FLI (kW·m$^{-1}$) (Finney et al., 2021; Rio et al., 2010).
    - Fire Radiative Energy (FRE, TJ): Satellite-measured energy emitted by the fire (TJ) allows us to obtain a directly measured heat flux. However, this measure is unreliable for low-intensity and small fires due to limitations in spatial and intensity resolution (Wooster et al., 2021).

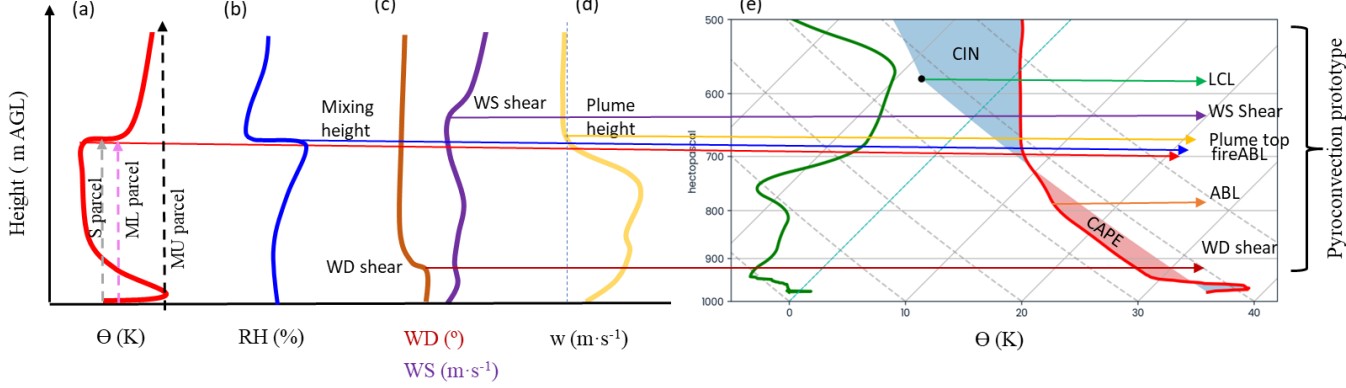

**Figure 3:** Characterizing wildfire dynamics with respect to ABL dynamics. A theoretical-sounding data representation is used to schematize the criteria for visually inspecting radiosounding variable profiles. a) The θ profile is used to obtain the potential parcel heights as a proxy of the plume height. We show ML, MU, and S parcels, initialized using the layer-averaged θ at the bottom 150 hPa of the captured vertical profile, the maximum θ at the same 150 hPa layer, and the surface θ, respectively. b) The RH profile is used to assess the mixing layer height in the plume area. c) Wind direction (WD) and speed (WS) shear are represented to the level of the highest gradient, d) w profile or rise velocity profile is used to assess the plume top when the updraft rise velocity returns to the ambient value. The skew-T diagram using $T_d$ and $T_s$ is used to visually assess ABL and LCL heights. The proximity of the turbulence levels LCL, ABL, fireABL, WS shear, and WD shear are used to assess the pyroconvection prototype (Castellnou et al., 2022).

**2.6 Characterizing ABL dynamics**

The criteria used to characterize wildfire dynamics relative to ABL dynamics (Figure 3) combine numerical estimates based on physical criteria with visual inspections of plotted profiles.

- ABL: The height of the maximum RH value is used as a criterion to estimate the height of the atmospheric boundary layer. This criterion is based on the observation that specific humidity tends to be well-mixed in the convective boundary layer (CBL), a condition conducive to fire spread. In this layer, temperature decreases with height, leading to an increase in relative humidity (RH) with altitude, reaching a peak at the inversion point. Above this inversion, the air becomes drier and warmer, resulting in a decrease in RH (Li et al., 2021; van Stratum et al., 2014; Vilà-Guerau de Arellano et al., 2015).

    We differentiate between the ambient ABL and the updraft fireABL. The latter refers to the thermodynamic changes in the ABL induced by the fire plume and restricted to the plume's area. The fireABL height is often identified by the plume injection height (Castellnou et al., 2022; Moisseeva, 2020).

    Numerically (S6), the ABL and the fireABL height are complemented by computing them using the bulk Richardson number (Rib) based on the ambient and in-plume profiles (Zhang et al., 2014).

- Wind shear: The height of the maximum gradient in the wind speed ($s^{-1}$) and direction profiles.

- Measured Plume top: The height at which the rising velocity of the in-plume sonde stabilizes back to the ambient sonde values. Due to the difference in density between helium and air, a sonde in the ambient average lower troposphere is expected to rise at 2 m·s$^{-1}$.

- Maximum potential plume top: The height to which the air parcel may rise using the parcel method (Holzworth, 1964; Seibert et al., 2000). Different parcel definitions following earlier pyrocloud studies (Lareau and Clements, 2016; Tory et al., 2018), initialized at launching height above ground level (AGL, m) and assuming dry adiabatic ascent. These include the most unstable (MU), the mixing layer (ML), and surface (S) parcels. The MU parcel uses the highest temperature value from the bottom 150 hPa of the vertical profile. The ML parcel uses the mean temperature and mixing ratio within the same 150 hPa layer. The S parcel reflects the surface temperature trajectory, initialized with a surface value of +3K (Luderer et al., 2009; Potter, 2005).

- LCL: In the Skew-T diagram, the LCL is identified at the pressure level where a parcel rising dry adiabatically from the surface temperature intersects the mixing ratio line associated with the surface dew point temperature. The LCL is computed numerically based on surface values using the METPY library (May et al., 2022). A direct estimation (Appendix S6) can also be provided using surface and dew-point temperatures (Bolton, 1980; Romps, 2017).

- CAPE / CIN: The integral of the differences between the theoretical undiluted parcel ascent trajectory (parcel method) and the ambient Ts profile. When plotted, CAPE or convective available potential energy is visually estimated as the area where the Ts parcel trajectory > Ts ambient profile, otherwise, the convection is inhibited (CIN). In this study, we examine how the air parcel in the fire front ascends at a higher temperature and humidity than its surroundings values, considering the level of free convection at the surface (Jenkins, 2004).

**Table 2:** Data, observations, and sources used for in-situ and real-time plume pyroconvection prototype analysis. The ambient and updraft sonde profile observations serve as the data source for visual estimates of levels and parcel trajectories along the state variable's graphical profile. Information about fire behavior is obtained from the fire service. Meteorological radar measurements are sourced from the Catalan Meteorological Service, when available. Additionally, complementary heat flux measurements are gathered from geostationary satellites.

|  | Variable | Description | Units | Source |
|---|---|---|---|---|
| Readings | sonde ascending profile | Track of the radiosonde path horizontally and vertically. | UTM, m AGL | Profile observation |
|  | $T^a$ ($T_s$, $T_d$) | Absolute temperature | K | Profile observation |
|  | RH | Relative humidity | % | Profile observation |
|  | P | Pressure | hPa | Profile observation |

| Category | Symbol / Name | Description | Units | Method / Source |
|---|---|---|---|---|
| | U | wind speed | m·s⁻¹ | Profile observation |
| | w component | | m·s⁻¹ | Profile observation |
| Variables (S6) | u component | | m·s⁻¹ | Computed from profile observation |
| | v component | Vertical wind speed | m·s⁻¹ | Computed from profile observation |
| | $q$ | specific humidity | g·kg⁻¹ | Computed from profile observation |
| | $\theta$ | potential temperature | K | Computed from profile observation |
| | $\theta_v$ | Virtual potential temperature | | Computed from profile observation |
| Fire-atmosphere interaction (S6 for alternative equations) | Measured plume height | | m | Visually displayed on the profile: rise-speed sonde profile stability |
| | | | | Radar echotop filtered at 12dBZ |
| | Potential plume height | Plume height estimated by the different parcel methods | m | Parcel method (see parcels type below) |
| | LCL | Lifting Condensation Level, Height at which a parcel of moist air lifted dry-adiabatically would become saturated | m | Visually displayed on the Skew-T |
| | ABL | Atmospheric Boundary Layer | m | Visually displayed on the profile: Maximum RH on the ambient sonde profile |
| | fireABL | fire-induced ABL. Modified mixing layer by plume turbulence mixing in the plume area and below the plume umbrella | m | Visually displayed on the profile: Maximum RH value on the in-plume sonde profile |
| | Wind shear | Wind direction and wind speed vertical gradient | s⁻¹ | Visually displayed on the wind speed profile |
| | CAPE / CIN | convective available potential energy / Convective inhibition | J·kg⁻¹ | Visually displayed on the Skew-T diagram |
| Parcels | S | surface parcel | K | $T_s$ at the surface |
| | ML | mixing layer parcel | K | $T_s$ averaged at lower 150 hPa |
| | MU | most unstable parcel | K | Maximum $T_s$ at lower 150 hPa |
| Fire | FRP | fire radiative power | TJ | Obtained from geostationary satellites |
| | FLI | Expresses the energy the fire is releasing per unit of the forward spreading front | kW·m⁻¹ | Obtained from measurements by the fire service |
| | Heat per unit area | Expresses the energy the fire is releasing per unit of surface in the flaming front | kW·m⁻² | Obtained from measurements by the fire service |
| | hourly isochrones | Hourly perimeter increment by the observed fire spread | ha | Obtained from measurements by the fire service |
| | Fuel type | Types of vegetation spreading the fire | Fuel model | Scott&Burgan general models: GR (grass), SH (shrub), TU (shrub under trees), TL (litter under tree) |
| | ROS | Fire front rate of spread | m·s⁻¹ | Obtained from measurements by the fire service |
| | Altitude | Fire front altitude above sea level | m ASL | Sonde launching points |
| | Coordinates | Fire front location | UTM | Sonde launching points |
| Plume | indraft | radial surface wind at the smoke plume base induced by an updraft | m·s⁻¹ | Profile observation |
| | updraft | rising convective wind inside a smoke plume. it is the in-plume w component | m·s⁻¹ | Profile observation |
| | umbrella | The thick smoke layer downwind from the head fire | m AGL | Profile observation |
| | overshooting | Dry turbulence rising above the average plume top and umbrella. | m | Profile observation |
| | pyroCu | Cloud formed by a rising thermal from a fire when it reaches LCL (American Meteorological Society, 2021). | | See Table 3 |
| | pyroCb | Extreme manifestation of a pyroCu when deepening above LCL and rising to the upper troposphere or lower stratosphere (American Meteorological Society, 2021). | | See Table 3 |

## 2.7 Pyroconvection prototype assessment

EWE are typically distinguished between dry convection and the moist convection, driven by the deep plumes that form pyroclouds (Rothermel, 1991). Pyroclouds types include shallow pyroCu, towering pyroCu, and intense pyroCb (Peterson et al., 2017). By examining ABL dynamics and the plume top position relative to ABL, LCL, and wind shear height (Castellnou
300  et al., 2022), we define six different plume prototypes or regimens (Table 3): those driven by dry convection: surface plume, convective plume, overshooting pyroCu, and those driven by moist convection: shallow pyroCu, towering pyroCu, and PyroCb.

Fire-atmosphere interaction can alter the vertical profile, potentially triggering a transition between different pyroconvective prototypes. Comparing ambient with in-plume state-variable profiles, aids in identifying potential pyroconvection prototype
305  transitions.

For clarity, based on Table 3 criteria, the example illustrated in Figure 3 will be classified as a dry convective plume due to the LCL/ABL ratio >>> 1 and the wind shear away from the ABL top. This profile indicates that no transition is possible now, for the plume below the Ɵ inversion and at a significant distance from the LCL level.

310  **Table 3:** Definition of pyroconvection prototypes. By examining the relative position of a plume concerning the ABL, LCL, and wind shear, we can identify different pyroconvective prototypes (Castellnou et al., 2022). We provide a brief description, by prototype, of the plume characteristics and their effects on fire spread relative to previous fire behavior.

| | Pyroconvective Prototype | Plume top height | LCL/ABL height ratio | Windshear height | Plume Description | Effect on fire spread |
|---|---|---|---|---|---|---|
| **Dry pyroconvection** | **Surface plume** | < ABL | >>>1 | Away from ABL top | Plume diluted inside the ABL | |
| | **Convective plume** | =>ABL | >>1 | Above ABL top | Plume reaching the ABL top and/or overshooting into the free troposphere (FT) | Fire behavior intensification and short-distance spotting |
| | **Overshooting pyroCu** 'opyroCu' | >ABL | >1 | ABL top but below LCL | Plume reaching the FT but limited by wind shear. They create short-living pyroCu pulses | Sustained fire spread acceleration, and constant short-distance spotting. Perimeter elongation |
| **Moist pyroconvection** | **Shallow pyrocumulus** 'Shallow PyroCu' | >ABL | =<1 | ABL top but on top of LCL | Plume reaching LCL, and forming pyroCu but limited by stability or wind shear in the FT | Sustained fire spread acceleration. Perimeter expansion pulses. Long-distance spotting |
| | **Towering pyrocumulus** 'Towering PyroCu' | >>ABL | =<1 | coinciding with ABL top and LCL | Plume reaching LCL and forming a deep pyroCu, NOT reaching Tᵃ < -35ºC | Sustained extreme spread and possible downdraft expanding chaotic fire |
| | **Pyrocumulonimbus** 'pyroCb' | >>>ABL | =<1 | coinciding with ABL top and LCL | Plume reaching LCL and forming a deep pyroCu with Tᵃ < -35ºC | Sustained chaotic expanding fire behavior, due to downdraft, sustained long-distance spotting |

## 3 Results of In-situ plume measurements and assessment of pyroconvection potential

We structure the results section as follows: first, we analyze the differences between the atmospheric model profile and the
315  observed in-situ ambient and in-plume updraft radiosonde profiles. Next, we compare measured state variable profiles of updrafts and ambient conditions to evaluate how well small balloon sondes detect changes from fire-atmosphere interactions and identify plume tops. Finally, we assess the sensitivity of this analysis to different convection conditions, focusing on updraft-launching positions and scenarios with multiple sondes, particularly regarding pyroconvection regime prototypes as detailed in Table 3.

**3.1 Atmospheric models profile compared with small ballons ambient and in-plume radiosonde profiling**

Figure 4 compares the ICON-EU model profiles with in-situ ambient and in-plume profiles of thermodynamic variables for two early-stage wildfires: Granja d'Escarp (118 ha) and La Selva de Camp (3.2 ha). We use a Skew-T diagram and the S parcel method to evaluate plume ascent relative to the LCL (black dot) and visualize CAPE (red shadow) and CIN (blue shadow). We aim to validate in-situ measurements using small balloons to effectively provide detailed profile measurements for assessing pyroconvection conditions.

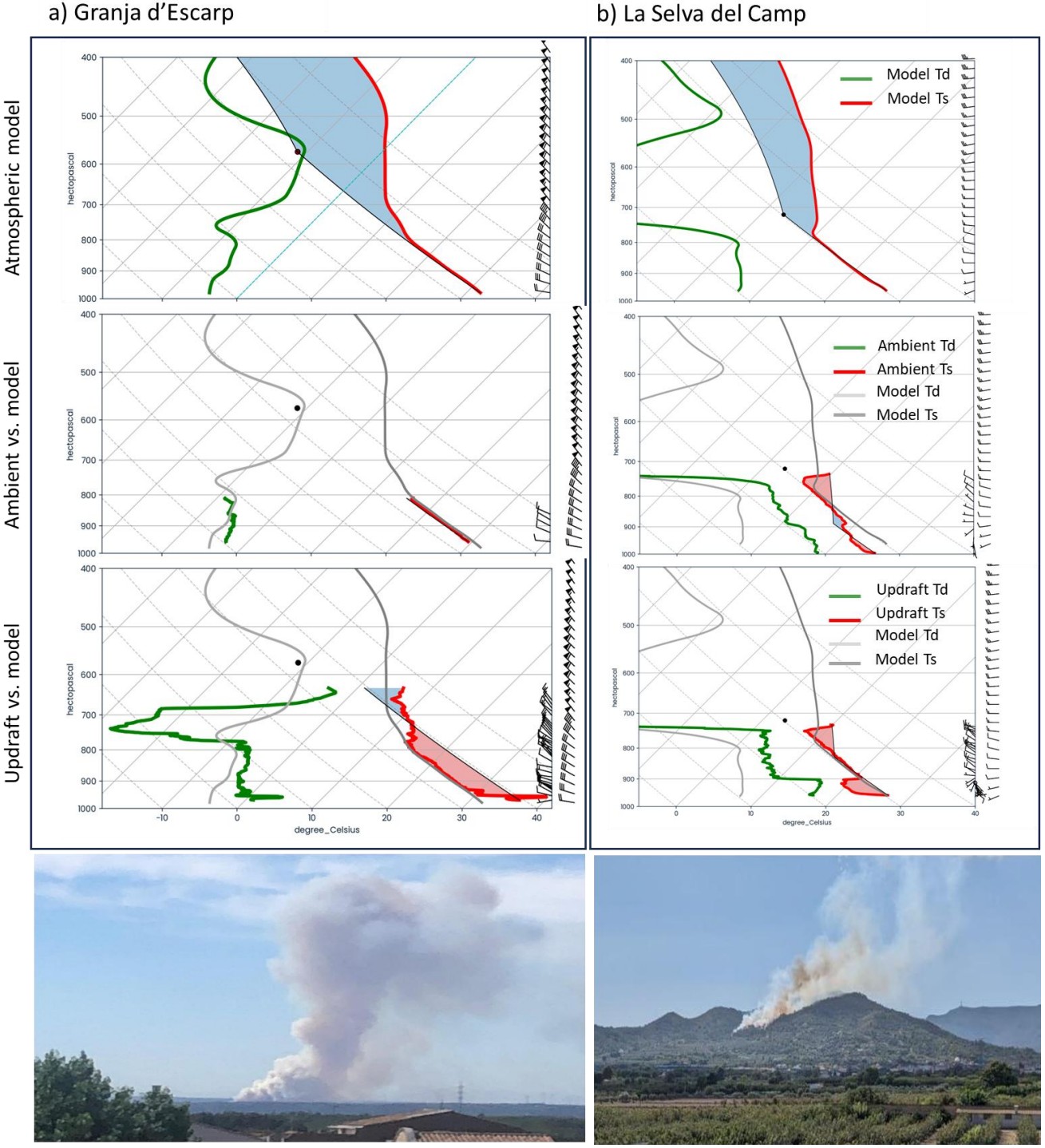

| Fire event / type indraft | Region | Fuel | in-plume hour (UTC) | Ambient hour (UTC) | area (ha), (Total/hour) | FLI(kW· m⁻¹) | ROS (m·s⁻¹) | FRE (TJ) | Prototype |
|---|---|---|---|---|---|---|---|---|---|
| Granja d'Escarp | ME | TU5 | 16:37 | 16:58 | 118 / 36 | 26741 | 1.05 | 2.1 | Convective |

| 03-07-2024 Head indraft | The sonde validates the model forecast and shows no transition to pyroconvection. | | | | | | | |
|---|---|---|---|---|---|---|---|---|
| La Selva del Camp 03-08-2023 Rear-indraft | ME | SH5 | 15:33 | 15:50 | 3.2 / 0.09 | 1258 | 0.029 | n. d. | Surface |
| | Sonde detected unexpected pyroCu potential not predicted by the model, leading to a safety debriefing for firefighters and a shift in tactical priorities. | | | | | | | |

**Figure 4:** Comparison of atmospheric model with in-situ ambient and updraft sonde vertical profiles for high and low-intensity fires in Catalonia (Spain). Additional fire information is available in Table S1. **Panel (a)**: La Granja Escarp fire (118 hectares) on July 3, 2024. The ICON-EU atmospheric model profile is shown at 17:00 UTC, the ambient sonde was launched at 16:48 UTC, and the updraft sonde at 16:37 UTC. **Panel (b)** La Selva del Camp fire (3.2 hectares) on August 3, 2023. The ICON-EU model profile is presented at 15:00 UTC, the ambient sonde was launched at 15:50 UTC, and the updraft sonde at 15:33 UTC. In the Skew-T diagram, we indicate the S parcel method, which also illustrates the Convective Available Potential Energy (CAPE, shown in red shading) and its inhibition (CIN, depicted in light blue shading). **Bottom table**: Region, total and current-hour burnt area (ha), launching hour for in-plume and ambient sonde, Fireline intensity (FLI, kW·m$^{-1}$), rate of spread (ROS, m·s$^{-1}$), heat flux captured by satellite infrared sensors (FRE, TJ), and the pyroconvection prototype as in Table 3.

Focusing first on the high-intensity La Granja d'Escarp fire (Figure 4a), the model profile categorized the fire as a convective plume prototype, with a CBL reaching up to 810 hPa and an LCL at 580 hPa. The ambient sounding showed similar characteristics for the CBL and LCL, but it was unable to capture data above the CBL because the sonde drifted away with the sustained winds. The head-infraft in-plume sonde observed marked differences with respect to the two other profiles. At the surface, it recorded an excess of 8 °C and a slightly moister profile. However, a significantly drier layer was identified between 790 and 690 hPa. Ascending inside the plume, the updraft sonde was able to ascend higher than the ambient sonde. On such a profile, the parcel method suggests a potential height of up to 720 hPa and indicates a significant fire-CAPE (red shadow). Despite potential parcel ascents, the measured updraft temperature difference relative to the ambient air decreased to below 1 °C at 980 hPa and was completely diluted at 790 hPa. The updraft readings suggest no expected change in the convective plume prototype among the three profiles, as reaching the LCL remains unachievable despite the enhanced convective plume.

The La Selva del Camp fire (Figure 4b) was a low-intensity fire spreading downhill, with a rapidly diluting surface plume. However, the presence of shallow cumulus clouds associated with sea-breeze advection prompted firefighters to be concerned about the potential transition into a pyroCu prototype. The absence of local sea breeze advection in the atmospheric model profile accounts for the significant discrepancies observed between the atmospheric model and in-situ profile measurements. The atmospheric model profile indicates a deep convective plume extending to 790 hPa, suggesting that further ascent is inhibited, making the LCL at 720 hPa difficult to reach. In contrast, the ambient sonde detects the local sea breeze, characterized by a specific humidity of 7 g·kg$^{-1}$, with the LCL now at 880 hPa. This suggests that a shallow pyroCu prototype could be triggered by an increase in S parcel temperature by 3 K. Such a temperature increase would overcome the minor convective inhibition (CIN) at 910 hPa and extend a pyroCu up to 730 hPa. The in-plume profile shows a 4°C temperature increase but results in a weak, rapidly diluted surface updraft at 950 hPa. This leads to a plume constrained below the 900 hPa inversion and with the LCL at 820 hPa. The fire's updraft was too weak to reach the LCL, despite an absence of CIN in the theoretical parcel trajectory. Our measurements confirm that transitioning from the weak plume to a shallow pyroCu prototype is possible but unlikely, requiring significant changes in fire behavior to strengthen updrafts, which is challenging under the current conditions.

The two examples in Figure 4 represent the additional value of in-situ profile observations, which can be used to adjust the maximum pyroconvection conditions possible by using the parcel method on in-situ plume updraft profiles.

In both high and low-intensity cases, the fire-induced updraft temperature drops quickly below 950 hPa, deviating from the expected maximum parcel ascent. This creates uncertainty about the plume top's location, hindering our understanding of pyroconvective conditions in relation to the ABL and LCL (Table 3). Locating the true diluted plume top becomes essential for a more accurate assessment.

## 3.2 Using updraft radiosondes for measuring plume top heigh and plume state variables

To assess the height of the plume top, it is crucial to determine whether the updraft is lifting the sonde. In alignment with the key variables used in detailed wildfire plume models (Freitas et al., 2007; Rio et al., 2010), we focus on how the fire alters the measured profiles of virtual potential temperature ($\theta_v$) and the rising velocity (w).

In Figure 5, we display the profile of differences in virtual potential temperature ($\theta_v$) between paired updraft and ambient sondes measurements. We analyze the sensitivity of these $\theta_v$ differences profiles to determine the plume top. First, we combine all the in-plume soundings (Figure 5a) and then separate the $\theta_v$ differences profiles by each indraft launch position (Figure 2). To allow a more systematic comparison, the height for each sonde has been normalized using the height at which the sonde profile equals ambient values of both $\theta_v$ and rising velocity, assuming this represents the plume well-mixed fireABL. When analyzing the differences between all paired updrafts-ambient sondes trajectories (Figure 5a), we observed the expected $\theta_v$ excess within a plume updraft. This effect is particularly noticeable in the flank and especially in the head of the indraft profiles (Figures 5c and 5d). However, this increase is quickly diluted beyond 50% of the profile height.

Interestingly, we observe some negative $\theta_v$ differences. This may be explained by the trajectory of a single sonde passing through the turbulent nature of an updraft, which entrains air from the surrounding atmosphere and by evaporation processes related to moisture from burning vegetation. The rear indraft (Figure 4b) shows more instances of no differences or negative differences compared to flank and head indraft values during its first 50% of the profile height. This rear indraft location corresponds to the main indraft flow into a fire head (Finney et al., 2015).
.

Overall, the differences in the updraft-ambient θv profiles show expected temperature increases in an updraft. However, rapid dilution to cooler values indicates that θv is not conserved due to entrainment of colder air. Thus, θv is not a reliable variable for assessing updraft height or identifying plume tops.

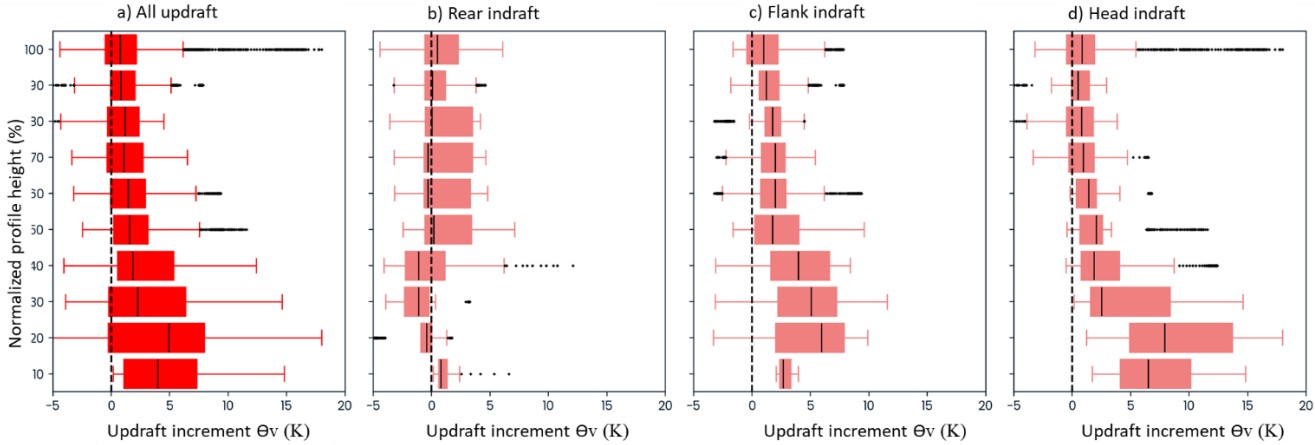

**Figure 5:** Patterns of differences between updraft and ambient profiles for virtual potential temperature $\theta_v$ (K). To facilitate intercomparison, each sonde height profile has been normalized with the height when the updraft sonde returns to the ambient sonde value (Castellnou et al. 2022). The zero difference is marked with a vertical black dashed line. We consider four profile types: generic updraft (all updraft profiles), the rear indraft, the flank indraft, and the head indraft (See Figure 2). The temperature profiles are expressed as the difference between the updraft and the ambient profile. None of the updraft profiles differ from each other ($p > 0.005$)

Figure 6 compares the rising velocity profiles for the updraft sondes launched from various indraft positions with those of the collocated ambient sondes. The profile height is normalized, as in Figure 5.

The ambient sonde (Figure 6a) consistently indicates the expected average ascent speed of 2 m·s⁻¹ (red dashed vertical line). The most well-defined profile corresponds to the rear indraft (Figure 6b), which in Figure 5 was the profile with less $\theta_v$ difference between updraft and ambient values. This profile features a consistently accelerated rising velocity beyond 30% of the plume's height. The lower rising speed in the first 30% is due to the launch position being behind the plume. As a result,

the sonde travels nearly horizontally before ascending (see Figure 2). It is the most reliable observation for assessing a vertical profile. In contrast, the sonde on the flank indraft (Figure 6c) is the weakest. Such sondes often take less reliable paths and may only enter the plume at higher altitudes (Figure S4). They can get caught in rotating coherent structures, like horizontal rolling vortices (HRV), which form within intense convective plumes (Finney et al., 2021). Conversely, the head indraft profile accelerates rapidly in the lower section, up to 60% of the height, but then loses strength.

Notably, the flank, head, and rear indrafts (Figure 6b, c, and d) show an increase in rising velocity that distinguishes them from the ambient profile (p=0.005). It is important in defining our criteria that the indraft profiles match the ambient average rising velocity at 90% of their height. This confirms that the rising-velocity vertical profile is a valid criterion for differentiating between in-plume and ambient sondes and for identifying the plume top when both ambient and in-plume velocities are equal and stabilize around 2 m·s$^{-1}$.

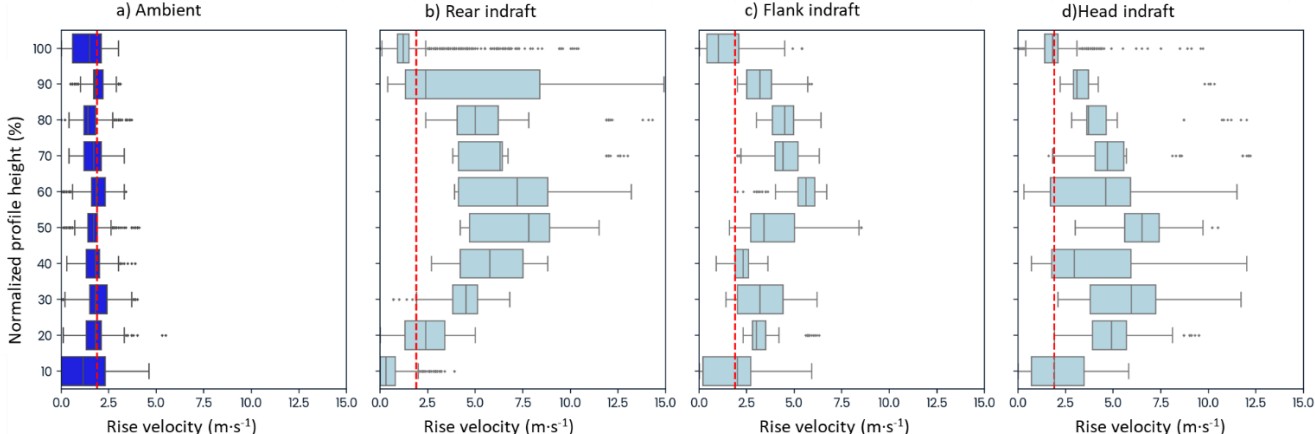

**Figure 6**: Patterns of rising velocity (m·s$^{-1}$) vertical profile observed in the updraft sondes. To facilitate intercomparison, each sonde height profile has been normalized with the maximum profile height without descending motion. We consider four profile types: The ambient, the rear indraft, the flank indraft, and the head indraft (See Figure 2). The average ambient sonde rising velocity (2 m·s-1) is marked as a dashed red line. All the updraft profiles are different from each other and from the ambient profile (p < 0.005). In addition, the profile at 90% of its height returns to the ambient average rising velocity, suggesting that vertical speed is an effective variable for detecting the plume top.

It is important to emphasize that within the first 10% of the height profile for the indraft sondes (Figure 6), the rising velocity is very similar to the ambient values. We analyze these initial ascent moments in Figure 7, comparing ambient and indraft sondes, as they are vital for validating the success of the launch early on. A detailed analysis in Figure 7a shows that updraft sondes ascend at the same velocities as ambient sondes in a layer from the surface to 200-300 meters. This pattern is consistent with profiles from ambient, dry, and moist pyroconvective plumes (Figure 7b). It resembles a layer at the plume neck where heat from the fire dissipates, unlike the upper layers where thermals actively organize heat transport (Rio et al., 2010).

Based on observational evidence, this layer serves as a guideline for distinguishing updraft radiosondes; beyond this point, the profile can be reliably distinguished from that of an ambient sonde.

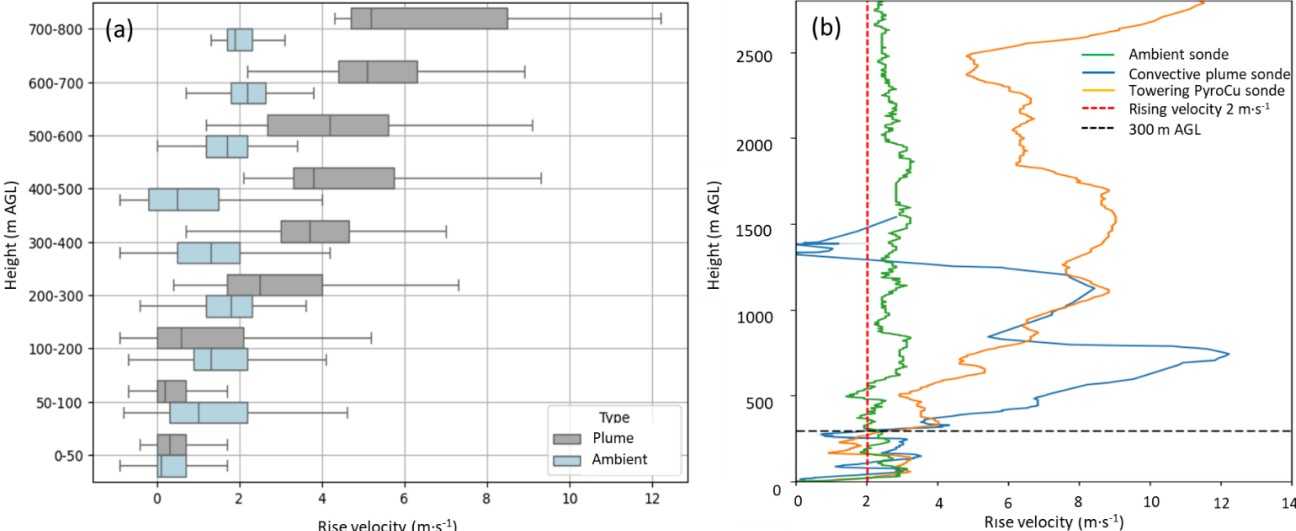

**Figure 7:** Observations of rising velocities to early differentiation between ambient and in-plume profiles. (a) Boxplot comparing the distribution of rising velocity by height classes (100 m intervals) for ambient (blue) and in-plume (grey) sondes. Ambient sondes maintain a rising velocity of 1.5 to 2 m·s⁻¹ on average, while in-plume sondes accelerate, showing a clear distinction from ambient sondes between 200 and 300 m AGL, with rising velocity exceeding, on average, 4 m·s⁻¹. (b) Detailed comparison of single sondes profile for ambient conditions (green), in-plume convective prototype (blue), and pyroCu prototype (orange). The three cases show a distinctive profile of rise velocity acceleration above the identified 300 m threshold.

Our findings in Figures 6 and 7 propose the rising velocity as a variable for a first-order estimate of the plume's top height. However, the uncertainty of a single sonde trajectory measure remains. In Figures S5 to S7, we present an uncertainty analysis of simultaneously launched sondes. This analysis demonstrates that using vertical velocity, along with relative humidity (RH) and virtual potential temperature ($\theta_v$), consistently identifies the maximum probability of plume top height, as shown by the distribution of plume top probabilities, with an averaged absolute error of 144 m. We reinforced the analysis using independent radar measurements for assessing whether the vertical velocity criteria defined in Figure 6 for estimating the dilution plume height is adequate. Figure 8 shows the correlation between the height at which the rising velocity of the updraft radiosonde returned to ambient values (radiosonde measured plume top) and meteorology radar echotops > 12dBZ (radar estimated plume top). Our dataset included 18 different fires, during which we launched in-plume radiosondes near meteorological radars in Catalonia. The results showed a strong correlation in all cases, with minimal variation in plume top height and a mean absolute error of 166.7 m. To complete the analysis, we provide detailed information on two specific radiosondes—one representing moist convection pyroCu and the other representing dry convection plume types (Figure 7b). It was observed that the ascent speed of the sondes decreased significantly (w < 2m·s⁻¹) as they approached the radar-determined plume top.

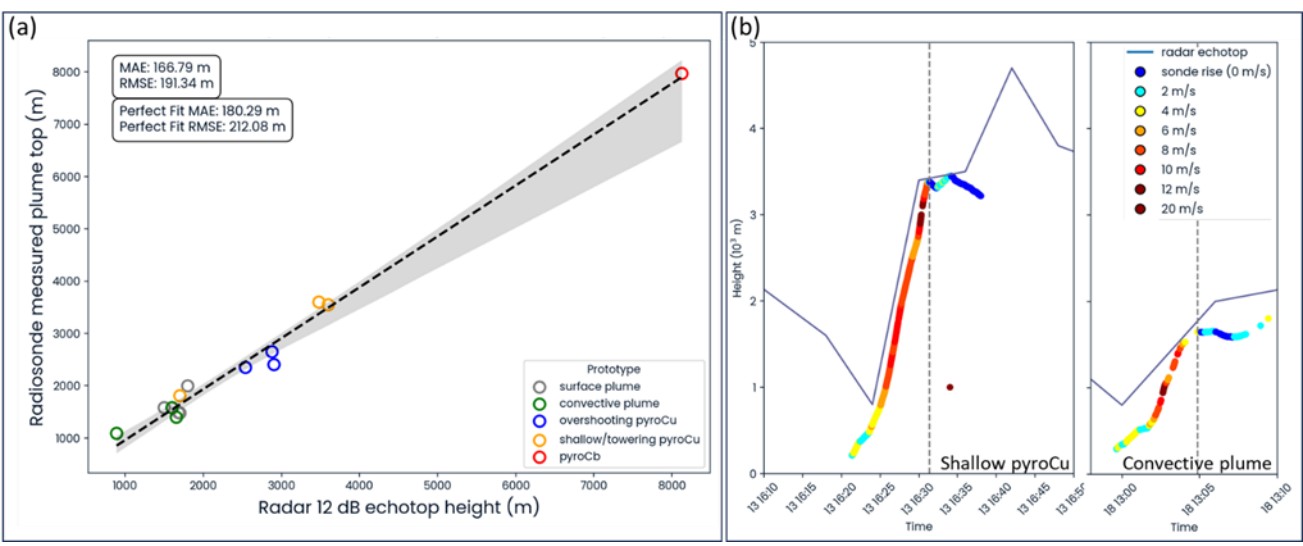

 **Figure 8:** Comparison of plume top height estimate using radar and soundings for different pyroconvection prototypes. a) Correlation of 18 sondes estimating plume top heights at the same hour and minute as the radar echotop readings at 12 dBZ. The mean absolute error (MAE) and the relative mean absolute error (RMSE) of the correlation indicate that the vertical profile of rising velocity from the sonde traveling within the plume updraft is a reliable proxy for estimating the plume's top height. b) We compare the radar echotop readings at 12 dBZ every 6 minutes with two different updraft sondes rising velocity profiles. The sonde updraft profile is colored to facilitate the reading of the rising velocity in m·s⁻¹. In both cases, the pyroCu and the convective plume prototype, we observe a close alignment between the sonde estimates and radar readings of the plume top.

## 3.3 Assessing pyroconvection transitions during ongoing operations

To evaluate the pyroconvection prototype, we compared ABL dynamics between collocated in-plume and ambient profiles (Table 3). We analyzed changes in parcel ascent using the $\theta v$ profile and assessed how the plume differs from the ambient mixed layer by examining relative humidity (RH). We also investigated wind direction and speed shear to determine if wind affects plume buoyancy and assessed the plume top by evaluating the updraft rise speed. Sensitivity analysis was performed using single sondes in dry and moist convective plumes with pyroCu/Cb, as well as multiple sondes within the same fire.

### 3.3.1 Dry pyroconvection prototypes

We compare low- and high-intensity fires with dry-convective plume prototypes (Figure 9). The Rojals fire in Catalonia , Spain (Figure 9a), is classified as a surface plume prototype within the ABL (Table 3). The updraft profile from a head indraft sonde shows a surface temperature excess of 7 K. However, this increase rapidly dissipates at 200 m AGL before reaching the 370 m AGL of the plume top identified by the rising velocity profile (dashed orange line). The plume does not deepen upon reaching the ABL top, as confirmed by the mixing layer height derived from the relative humidity profile (blue dashed line), which remains unchanged between the ambient air and the plume. This is further supported by varying wind shear values and higher wind speeds within the proposed plume height when compared to ambient values. In this scenario, the theoretical undiluted updraft height, estimated using the MU parcel method (black dashed arrow), is located at 1480 m AGL. This value is slightly lower than LCL but five times higher than the current diluted plume top. The ML parcel potential height (pink dashed line) is just above the diluted plume and mixing layer at 521 m AGL. We conclude that there will be no transition to a different pyroconvective prototype with these diluted updraft conditions.

The Santa Ana fire in Chile (Figure 9b) is categorized as convective plume prototype. The updraft profile from a head indraft sonde shows a temperature excess of 15 K. This temperature decreases to ambient values at 780 m AGL. The updraft rising velocity criteria estimates the plume top at 1317 m AGL. Such a measure is confirmed by both the RH profile (thick blue dashed line) proposing a fireABL deepening of 581 m above the ambient ABL (thin blue dashed line) and wind direction changes from the ambient ABL height to the suggested plume top. Notably, the rising velocity steadily decreases as the plume deepens in the stable region above the ambient ABL, where the plume $\theta v$ becomes up to 4K cooler than the ambient sonde. In contrast to the Rojals fire, this case provides evidence of a potential transition from a convective plume to an overshooting pyroCu prototype, as indicated by the MU parcel. Real-time observations showed that the plume top was close to the LCL, even though the fire's intensity was moderate at the time. This observation, along with firefighters' reports of an increasing rate of fire spread, alerted us to a possible sudden and dangerous change in fire behavior, catching the firefighters off guard. Thanks to the in-situ profiles, the crews were able to move to safety zones. Two hours later, the formation of a pyroCu confirmed the expected intensification of the fire. This aspect of the methodology is both unique and valuable, as it enables proactive tactical adjustments to enhance safety.

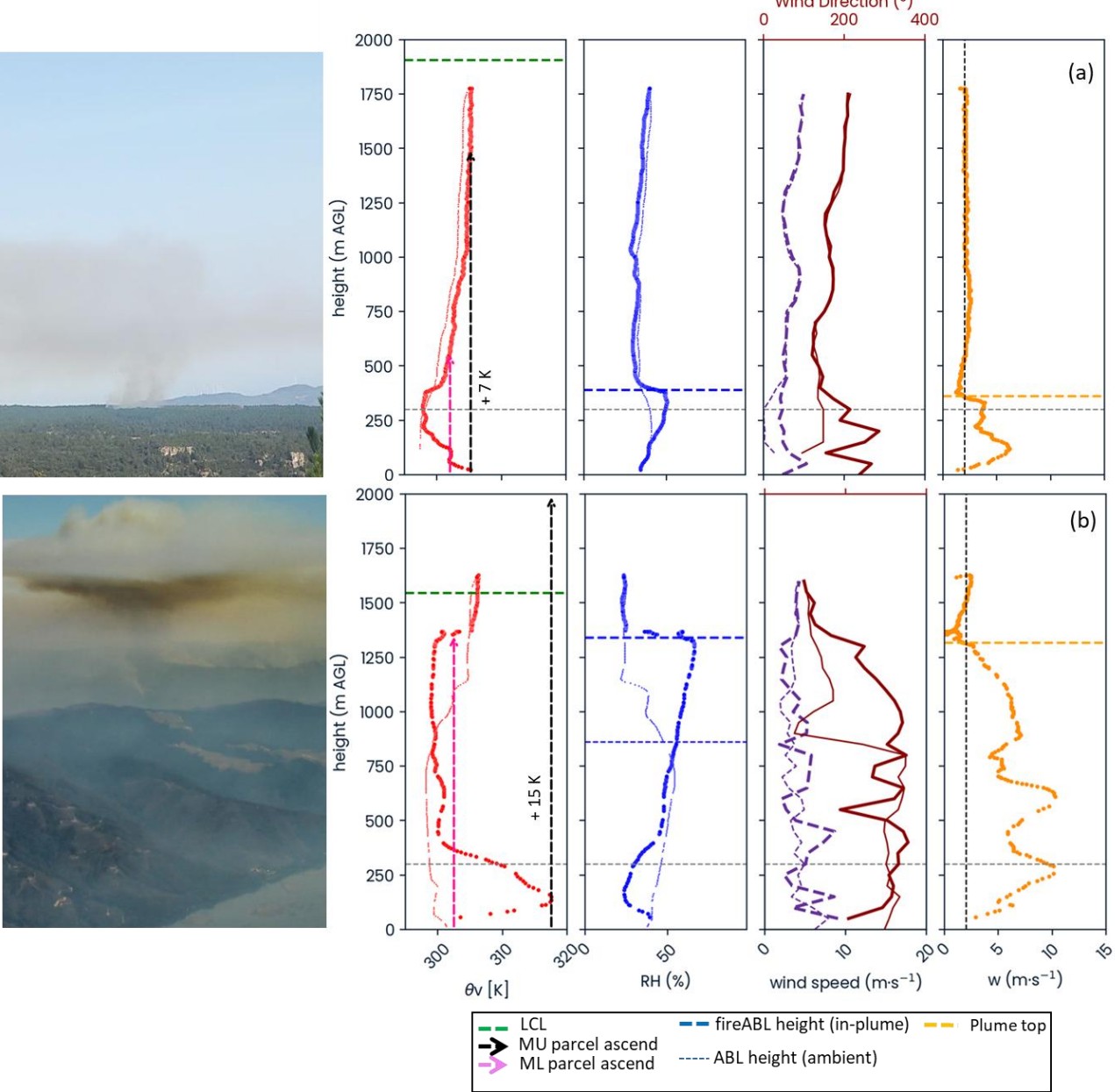

| Fire event/type indraft | Region | Fuel | in-plume hour (UTC) | Ambient hour (UTC) | area (ha), (Total / hour) | FLI (kW·m⁻¹) | ROS (m·s⁻¹) | FRE (TJ) | Prototype |
|---|---|---|---|---|---|---|---|---|---|
| Rojals 21-03-2024 Head indraft | ME | TL1 | 14:29 | 14:49 | 6 / 0.7 | 870 | 0.025 | n. d. | surface |
| | The sonde validates the model forecast and shows no transition to pyroconvection. | | | | | | | | |
| Santa Ana 19-02-2023 Head indraft | SA | TU5 | 17:3216:32 | 16:46 | 10412 / 736 | 28974 | 0.55 | 14 | Convective/ opyroCu |
| | Radiosonde indicates that pyroCu may form if the fire intensifies, prompting a recommendation for all firefighters to evacuate. Eventually, a pyroCu formed, and the fire accelerated. | | | | | | | | |

**Figure 9:** Interrelating state variable profiling for dry pyroconvection prototype. a) Rojals fire 2024, Catalonia (Spain). The updraft sonde is a head indraft type. The ambient sonde is launched 2.1 km to the W. b) Santa Ana fire 2023, Chile. The updraft sonde is a head indraft type. The ambient sonde is launched 4.5 km to the E.  Profiles use a thin line for the ambient sonde and a thick line for the updraft representing $\theta_v$ (K, red), Relative humidity (% blue), wind direction (º, violet), wind speed (m·s⁻¹, dark red) and vertical speed w (m·s-1, orange). The $\theta_v$ profiles include LCL (dashed green). The parcel method potential plume height by ML parcel (dashed pink vertical arrow) and MU parcel (dark dashed vertical arrow) is shown on the $\theta_v$. The RH (%) maximum value identifies the mixing layer height. The wind direction and speed profiles identify the wind shear. The rising velocity quantifies plume top heights (dashed orange line) using the 2m·s⁻¹ criteria. The horizontal thin and dashed grey line indicates the 300 m AGL needed to confirm an in-plume sonde, as in Figure 7.  Bottom table: Region, total and current hour burnt area (ha), launching hour for in-plume and ambient sonde, Fireline intensity (FLI, kW·m⁻¹), rate of spread (ROS, m·s⁻¹), heat flux captured by satellite infrared sensors (FRE, TJ), and the pyroconvection prototype as in Table 3.

### 3.3.2 Moist pyroconvection prototypes

In Figure 10, we examine the Martorell fire dynamics from 16:00 to 17:30 UTC. During this time, eight firefighters were trapped due to rapid changes in surface fire spread associated with the transition from a convective plume to a shallow pyroCu prototype.

At 16:29 UTC (Figure 10a), the in-plume vertical speed profile for the rear indraft sonde indicates a plume rising to 3430 m AGL, 800 m above LCL, producing a shallow pyroCu prototype. The rising velocity profile increases from 6 to 12 m·s-1 at 2620 m AGL, an effect attributable to latent heat release from pyroCu condensation (Rodriguez et al., 2020). The RH profile maintains high values (>90%) throughout the 1000m deep moist pyrocloud. Notably, the wind direction profile, initially with shear between 600 and 800 m AGL is changed by the plume to the the base of the pyrocloud. The resulting plume top by the pyroCu shows a plume height of 120% of the ML parcel expected height (pink dashed arrow), but similar to the MU parcel maximum potential height. Despite the fire's intensity and observed fireABL modifications, the $\theta_v$ profile shows minimal difference from the ambient profile, consistent with no $\theta_v$ excess on the rear indraft profiles (Figure 5). Based on the parcel analysis and the profile measured increase in stability and WS shear at the current plume top, the maximum pyroconvection prototype is likely achieved, and further deepening to pyroCb is unlikely.

Half an hour later, the same fire is spreading downhill at four times less intensity, forming a surface plume prototype. A head indraft sonde (Figure 10b), identifies the plume top at 1650 m AGL (orange dashed line) from a quick diluting updraft ($\theta_v$) from such descending front. The updraft is too weak to reach the LCL. However, the measured updraft $\theta_v$ profile with an excess of 8 K on the surface proposes a MU and ML ascent above LCL, pointing to a potential transition to a pyroCu prototype. It is important to be vigilant about this situation, as changes in fire spread could easily trigger the formation of a shallow pyroCu prototype, suddenly intensifying the rate of fire spread

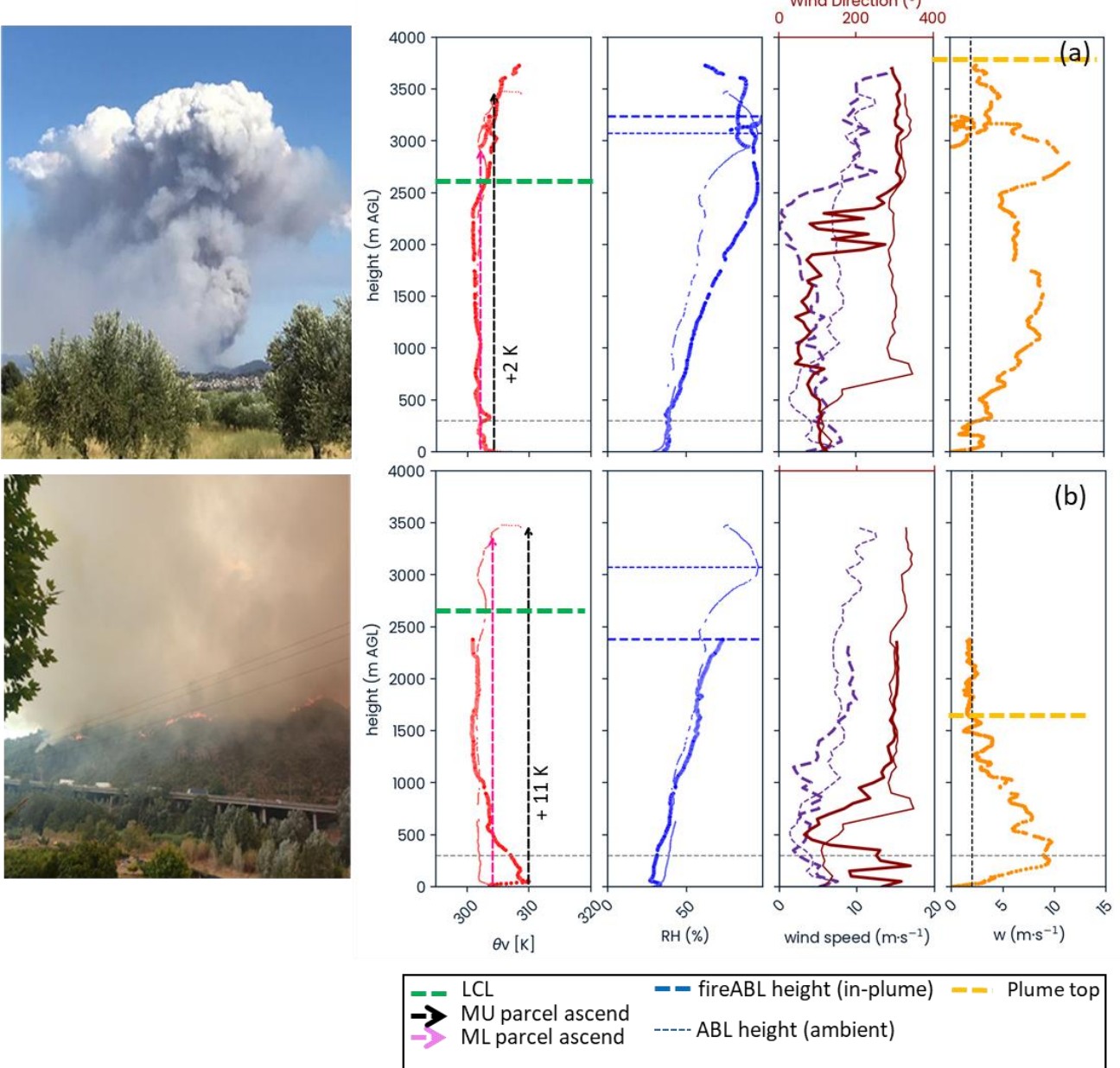

| Fire event/type indraft | Region | Fuel | in-plume hour (UTC) | Ambient hour (UTC) | area (ha), (Total / hour) | FLI (kW·m⁻¹) | ROS (m·s⁻¹) | FRE (TJ) | Prototype |
|---|---|---|---|---|---|---|---|---|---|
| Martorell 13-07-2021 Rear indraft | ME | TU5 | 16:29 | 16:42 | 298 / 123 | 12750 | 0.61 | 8.2 | pyroCu |
| | colspan | Evaluating the likelihood of a pyroCu transition to quantify the uncertainty of the situation. | | | | | | | |
| Martorell 13-07- 2021 Head indraft | ME | TU5 | 17:02 | 16:42 | 298 / 2.6 | 3000 | 0.083 | 1.1 | surface |
| | colspan | Assessing the possibility of a new pyroCu transition to measure the uncertainty of the situation. The radiosonde indicates a clear pyroCu transition if the fire intensifies. | | | | | | | |


**Figure 10:** Interrelating state variable profiling for moist pyroconvection prototypes during the Martorell 2021fire case in Catalonia (Spain)**.** a) Martorell fire at16:29 UTC, by a rear indraft sonde. b) Martorell fire at 17:02 UTC, by a head indraft sonde. The ambient sonde is launched 2,3 km to the SE at 16:42 UTC. Profiles use a thin line for the ambient sonde and a thick line for the updraft representing θ_v (K, red), Relative humidity (% blue), wind direction (º, violet), wind speed (m·s⁻¹, dark red) and vertical speed w (m·s-1, orange). The θ_v profiles
include LCL (dashed green). The parcel method for potential plume height by ML parcel (dashed pink vertical arrow) and MU parcel (dark dashed vertical arrow) is shown on the θ_v. The RH (%) maximum value identifies the mixing layer height. The wind direction and speed profiles identify the wind shear. The rising velocity quantifies plume top heights (dashed orange line) using the 2m·s⁻¹ criteria. The horizontal thin and dashed grey line indicates the 300 m AGL needed to confirm an in-plume sonde, as in Figure 7. Bottom table: Region, total and current hour burnt area (ha), launching hour for in-plume and ambient sonde, Fireline intensity (FLI, kW·m⁻¹), rate of spread (ROS, m·s⁻¹),
heat flux captured by satellite infrared sensors (FRE, TJ), and the observed pyroconvection prototype as in Table 3.

### 3.3.3 Multiple launching during ongoing operations in active wildfires.

A series of updrafts and ambient pairs of sondes were launched at the Casablanca III fire in Chile (Figures 11 and 12), which expanded to 12,073 hectares between February 8 and February 10, 2023. The situation was complex, as fires intensified on
February 2, burning over 362,000 hectares.

Figure 11 shows two sondes launched simultaneously from the same location (21:46 and 21:51 UTC) to try to assess plume height and deepening on top of the thick smoke layer (Figure 11a). The plume top could only be seen far upwind of the fire (Figure 11b), but not within the fire area.

The first sonde at 21:46 UTC shows an updraft excess of 22K on the $\theta_v$ profile. The plume top is identified by the rising
velocity profile at 2015 m AGL, confirmed by the RH proposed mixing layer (thick blue dashed line). The plume deepens 300 m above the ambient mixing layer (thin blue dashed line). The wind speed increases by 5 m·s$^{-1}$ in the plume between 300 and 1200 m AGL. The rising velocity has an average of 10 m·s-1 above 300 m until the same 1200 m AGL that wind speed is modified. From there to the top, it gradually loses strength. This section of the vertical profile coincides with the height where the plume updraft temperature is already diluted, and the plume rises above its neutral buoyancy level.

The second sonde, launched five minutes later, recorded an updraft excess of 23K, which dissipates quickly and at a lower altitude compared to the first sonde. The plume top is identified by the vertical velocity and RH profiles at 1880 m AGL. This time, the rising velocity is weaker, averaging 6.3 m·s$^{-1}$, and losing strength above the level of neutral buoyancy.

In both sondes, the MU parcel shows an unrealistic height potential if we do not account for the fast updraft temperature dilution. The ML parcel consistently provides a good assessment of the real plume height.

The plume height shows a difference of 225 m between the two measured plume top heights, indicating a plume slightly overshooting the ABL. The difference between simultaneous sondes shows a resolution according to the variance of a turbulent plume top, as observed by radar and satellite measurements (Lareau et al., 2024; Wilmot et al., 2022).


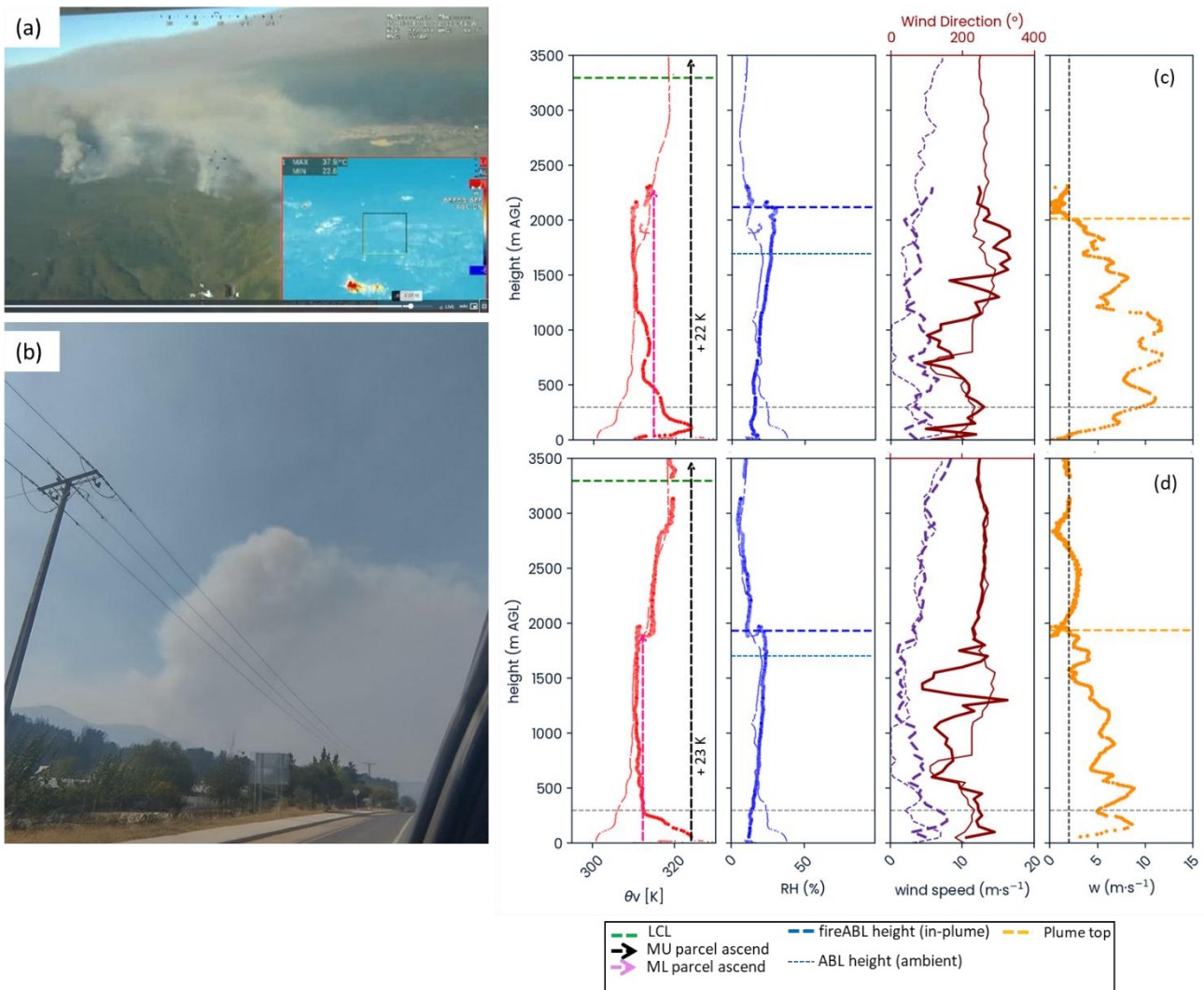

| Fire event/type indraft | Region | Fuel | in-plume hour (UTC) | Ambient hour (UTC) | area (ha), (Total/hour) | FLI (kW·m⁻¹) | ROS (m·s⁻¹) | FRE (TJ) | Prototype |
|---|---|---|---|---|---|---|---|---|---|
| Casablanca III 08-02-2023 Head indraft | SA | TU5/SH5 | 21:46 21:51 | 22:27 | 12073 / 1012 | 19720 | 0.85 | 11.5 | Convective |
| | The sonde confirms a scenario without transition, with an assessment of a persistent convective plume 1300 m below the LCL. | | | | | | | | |

**Figure 11**: Interrelating state variable profiling for two simultaneous sondes at Casablanca III fire the 8th of February. (a), The fire has a
thick smoke layer covering Chile's central Valley. (b). The plume of Casablanca fire can only be seen on the fringes of the thick smoke layer.
(c) Flank indraft sonde at 21:46 UTC. (d): Flank indraft sonde at 21:51 UTC. The ambient sonde is launched 4,8 km to the E at 22:27 UTC.
Profiles use a thin line for the ambient sonde and a thick line for the updraft representing $\theta_v$ (K, red), Relative humidity (% blue), wind
direction (º, violet), wind speed (m·s⁻¹, dark red), and vertical speed w (m·s⁻¹, orange). The $\theta_v$ profiles include LCL (dashed green). The
parcel method for potential plume height by ML parcel (dashed pink vertical arrow) and MU parcel (dark dashed vertical arrow) is shown
on the $\theta_v$. The RH (%) maximum value identifies the mixing layer height. The wind direction and speed profiles identify the wind shear. The
rising velocity quantifies plume top heights (dashed orange line) using the 2m·s⁻¹ criteria. The horizontal thin and dashed grey line indicates
the 300 m AGL needed to confirm an in-plume sonde, as in Figure 7.  Bottom table: Region, total and current hour burnt area (ha), launching
hour for in-plume and ambient sonde, Fireline intensity (FLI, kW·m⁻¹), rate of spread (ROS, m·s⁻¹), heat flux captured by satellite infrared
sensors (FRE, TJ), and the observed pyroconvection prototype as in Table 3.


On the 10th of February, the Casablanca III fire was already 8173 ha in size, and increased in size by 3900 ha. It was assessed

as a convective plume prototype without a pyroCu transition being possible due to an ambient LCL 2000 m higher than ABL

(Figure 12). Fire behavior was initially expected to calm in the early evening. However, the combined assessment of various

weather forecasts indicated a 60% chance of intensification during the transition from day to night. This potential increase in

fire activity is due to drier air moving from the southwest into the area. The important takeaways of the ambient-updraft sondes readings from the fireline are:

- At 20:03 UTC, the fire moves slowly after intense midday runs, with a shallow, diluted plume. A head indraft sonde profile shows a $\theta_v$ excess of 13 K, producing a potential MU parcel rise to the LCL (green line). A more realistic parcel rise is identified by ML parcel at 830 m AGL. The rising velocity profile, however, shows a plume top at just 896 m AGL, which coincides with the observed low plume strength. The RH and wind profile confirm no ABL height modification by the plume. The current conditions produce a weak surface plume prototype. However, the situation remains unstable, as enough fire spread can easily trigger a pyroconvection transition, as indicated by the MU parcel.

- At 20:29 UTC, a new flank updraft sonde confirms a much less intense $\theta_v$ excess of 5 K. The ML parcel now shows an 810 m AGL height, and MU reaches the potential height of 2700 m. The plume top of a weaker surface plume is measured at 615 m by the rising velocity, as confirmed by unmodified HR and wind speed profile between ambient and updraft conditions. The situation appears to be stabilizing; however, we must remain aware of the potential for a deeper plume, as suggested by the MU parcel.

- At 22:06 UTC, a reignition on the flank further south started a new intense run. A flank updraft sonde was launched, showing a $\theta_v$ excess profile of 16 K up to 1670 m AGL, and a rising velocity profile proposing plume top at 1910 m AGL. The new fire now has a deepening plume, with the top now triple the previous plume top. It remains just below the ML parcel, deepening by 1000 m above the ambient ABL. This extreme is confirmed by changes in the RH and wind speed profiles up to the proposed plume top. The opening of the left flank is building an intense head fire using drier conditions advected into the area: the ambient RH decreased rapidly from 20% at 20:03 UTC to 8% at 22:06 UTC. Such a scenario proposes a convective plume height just 1100 m below LCL. If the fire spread continues at its current pace, we can assess a potential transition to an overshooting pyroCu prototype, as indicated by the unconstrained MU parcel potential height.

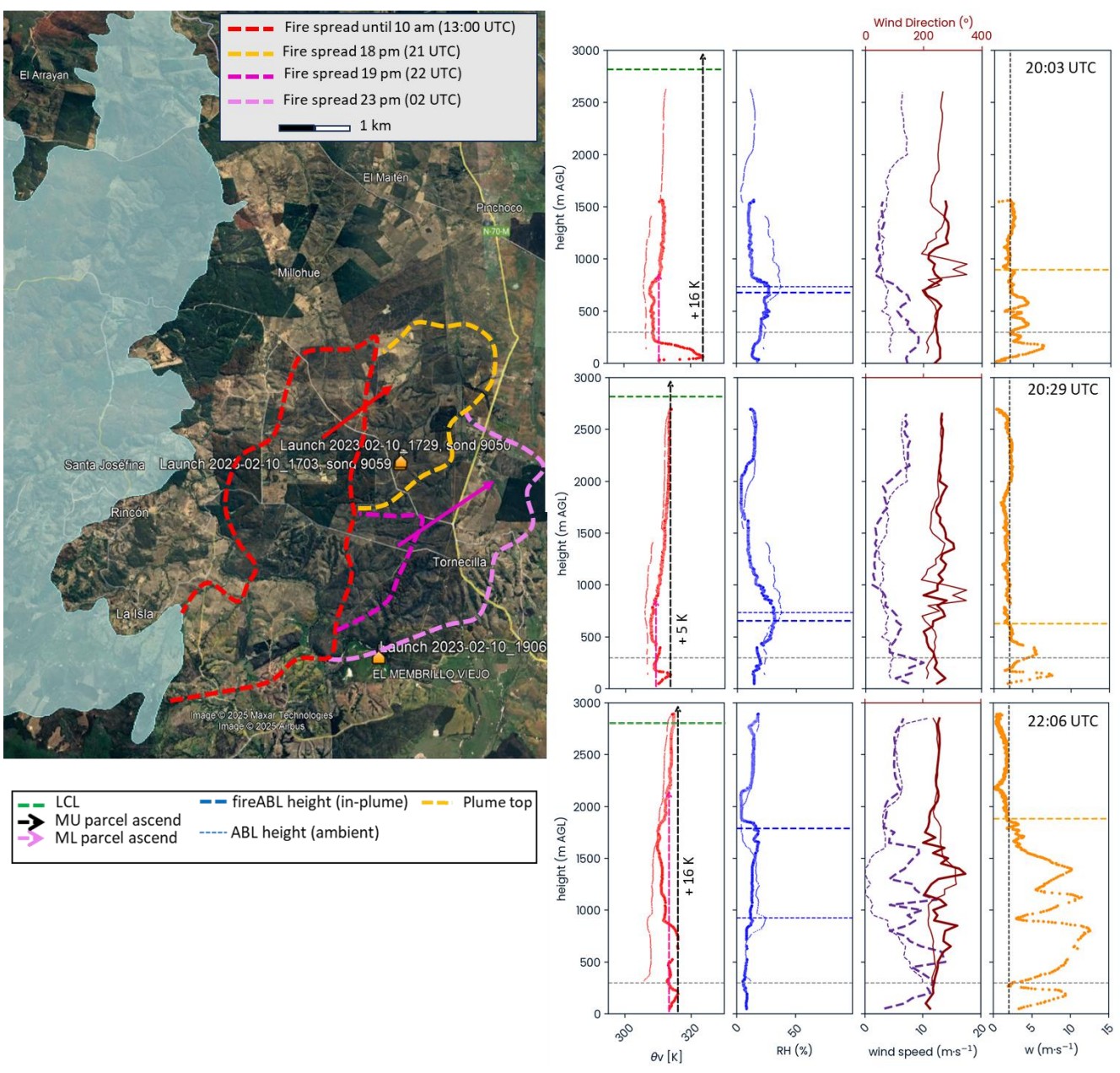

| Fire event/type indraft | Region | Fuel | in-plume hour (UTC) | Ambient hour (UTC) | area (ha), (Total / hour) | FLI (kW·m⁻¹) | ROS (m·s⁻¹) | FRE (TJ) | Prototype |
|---|---|---|---|---|---|---|---|---|---|
| Casablanca III 10-02-2023 Head indraft | SA | TU5/SH5 | 20:03 | 20:56 | 12073 / 11.6 | 2800 | 0.08 | 1.02 | Surface |
| Although a weak plume is diluting inside the ABL, the in-plume sonde confirms potential for the pyroCu transition. LACES protocol, focused on pyroconvection transition, is assigned | | | | | | | | | |
| Casablanca III 10-02-2023 Flank indraft | SA | TU5/SH5 | 20:29 | 20:56 | 12073 / 6.2 | 1756 | 0.02 | 1.1 | Surface |
| The LACES protocol on pyroconvection transition indicates calm conditions, but a transition to pyroCu remains possible if the fire front intensifies. | | | | | | | | | |
| Casablanca III 10-02-2023 Flank indraft | SA | TU5/SH5 | 22:06 | 22:30 | 12073/ 630 | 31114 | 1.1 | 7 | Convective |
| The intensification has resulted in a clear transition to pyroconvection, with the plume top now 1500 meters higher. A safety alert has been issued | | | | | | | | | |

610  **Figure 12:** Interrelating state variable profiling for the Casablanca III (Chile) fire, 10th February 2023. We show three updraft sondes (20:03 UTC, a head indraft, 20:29 UTC, and 22:06 UTC, a flank indraft) paired with two ambient sondes launched 7 km to the E at 20:56 and 22:30 UTC. Profiles use a thin line for the ambient sonde and a thick line for the updraft representing θ$_v$ (K, red), Relative humidity (% blue), wind direction (º, violet), wind speed (m·s-1, dark red) and vertical speed w (m·s-1, orange). The θ$_v$v profiles include LCL (dashed green). The

parcel method for potential plume height by ML parcel (dashed pink vertical arrow) and MU parcel (dark dashed vertical arrow) is shown on the $\theta_v$. The RH (%) maximum value identifies the mixing layer height. The wind direction and speed profiles identify the wind shear. The rising velocity quantifies plume top heights (dashed orange line) using the $2 m \cdot s^{-1}$ criteria. The horizontal thin and dashed grey line indicates the 300 m AGL needed to confirm an in-plume sonde as in Figure 7. Bottom table: Region, total and current hour burnt area (ha), launching hour for in-plume and ambient sonde, Fireline intensity (FLI, $kW \cdot m^{-1}$), rate of spread (ROS, $m \cdot s^{-1}$), heat flux captured by satellite infrared sensors (FRE, TJ), and the observed pyroconvection prototype as in Table 3.

## 3.4 Usability and Failure of Plume Profiling for Incident Management in Extreme Fire Events

Over the five years of fire campaigns, we obtained clear results supporting the use of paired ambient-in-plume profiling with radiosondes on active wildfires (Table 4). The low failure rate of 7.73% and the consistent application of sonde information for awareness improvement, tactical adjustments and safety decisions indicate that this methodology is well-suited for adapting operational tactics (73.27% of our case studies) to address the challenges posed by pyroconvection transitions and, in 13% of cases, to shut down operations and retire all firefighters to the safety zone.

It is important to note that during the campaigns, sondes that failed to enter the plume typically did so because they were launched too far from the plume base into weak or intermittent indrafts (Figure S4). This often happened in the head or flank indrafts. In contrast, sondes launched in the rear indraft needed to be launched far enough from the head fire to avoid being pushed to the ground by the descending flow of air into the plume neck. However, those sondes traveled farther when launched from the rear indraft into a strong indraft (Figure S9). This finding is particularly significant for extreme pyroconvective fires. Taller plumes generate stronger rear indrafts, which aid in the successful and safe deployment from the distance of rear indraft sondes into already established pyroconvective clouds (pyroCbs). In our campaigns, sondes were launched into pyroconvective bursts during the Santa Coloma Queralt fire in 2021 and the Guisona fire in 2025. These sondes were deployed kilometers behind the fire's leading edge and after traveling between 3 and 9 km in the indraft to reach the plume, finally successfully ascended into the pyrocloud, reaching altitudes exceeding 7000 meters AGL (Figures S8 and S9). This provides a clear opportunity for launching research sondes during extreme ongoing pyroCu/Cb events, as it is unsafe to remain near the front of the fire.

Table 4.- Summary of success and failure (and reason of failure) along with use in decision making of the sondes launched (Table S1).

| Type of sonde | Proportion over total sondes | description | |
|---|---|---|---|
| Failed sondes | 7.73% | 61.3% too weak indraft, or launching too far away | |
| | | 23% pushed to the ground by rear indraft | |
| | | 15.3% sonde failure | |
| Operational | 73.27% | Awareness | 51% |
| | | Tactical | 36% |
| | | Safety | 13% |
| Research | 19% | | |

## 4 Discussion

This methodology allows for the safe, systematic collection of ambient and in-plume measurements during wildfires, including both growing and extreme events. It focuses on assessing changes in state variables induced by plumes relative to ambient conditions, thereby improving our understanding of pyroconvection. Firefighters can utilize this approach to evaluate potential pyroconvection transitions in real-time. Further details on sonde placement and the underlying physics are provided below.

### 4.1 Small balloon's reliability for capturing local profile characteristics.

In-situ ambient and updraft profiles measured with operational small balloons effectively reduced uncertainty from atmospheric model resolution by capturing local singularities that those models cannot account for (Dutra et al., 2021; Salvador et al., 2016; Wagner et al., 2014). Across our campaigns, profiles with humidity advection were more prone to local singularities than those with dry convection or stable profiles (Figure 3).

The novelty of our methodology lies in systematically pairing ambient and updraft profiles. They provide real-time measurements to understand how fire-atmosphere interactions are altering the ambient ABL thermodynamics. These profiles directly measure the plume current $\theta$ and q values for the parcel potential rise, reducing the need to apply theoretical adjustments (Luderer et al., 2009; Potter, 2005). They also complement state-of-the-art methods (Artés et al., 2022; Leach and Gibson, 2021; Tory and Kepert, 2021).

However, it is important to note that with such small balloons, ambient sondes may not capture the full extent of the vertical profile in the presence of deep stability, subsidence, or wind speed shear. In these situations, the sondes tend to stabilize their ascent at the plume injection height layer. While it provides the necessary information, atmospheric models are sometimes needed to supplement data from above those layers (Eghdami et al., 2023).

## 4.2 Evaluating sonde data for capturing updraft variables and plume top

Comparing state variables between updraft and ambient conditions helps identify plume-induced changes in the ambient conditions, assess plume height, and raise awareness of current pyroconvection conditions.

The $\theta_v$, despite its use in estimating updraft potential maximum height by the parcel method, is not sensitive enough to identify the plume top. The temperature increase in the updraft is quickly diluted before reaching 50% of the plume profile, coinciding with previous measurements (Charland and Clements, 2013; Kiefer et al., 2009). Above this height, non-buoyant deepening is driven by plume mass flux momentum (Moisseeva, 2020). The near-zero or even negative values from the updraft-ambient comparison cause this variable to poorly identify the plume top without the other variable's profile assessment. The dry convection case at Santa Ana (Figure 9b) exemplifies this situation with -4 K the last 500 m of updraft rising between 900 and 1400 m AGL. The $\theta_v$ profile from the updraft deepening above the ambient ABL profile is 5 K cooler than ambient, resulting in an incorrect plume-top assessment by 580 m.

The updraft rising velocity profile helps us to better determine the plume top by identifying where plume dilution occurs. These profiles differ significantly from ambient rising velocity profiles. Additionally, radar plume top assessment data confirms that the updraft sondes, despite single trajectories that may not enter strictly within the internal cores, travel with the updraft, and ascend towards the top of the plume (Figure 8).

The measured rising velocity values of the updraft sondes, which reach up to 21 m·s⁻¹ (Figure 6), are lower than the extreme updrafts of deep pyroconvective clouds (pyroCb) observed by Doppler radar, where peak velocities range between 30 and 60 m·s⁻¹ (Lareau et al., 2024; Rodriguez et al., 2020). It is important to note that these measurements are localized to a small section of a significantly large pyroCb plume that extends to heights above 10-12 km, with average updrafts ranging from 8 to 19 m·s⁻¹. Other radar measurements reported rising-velocity peaks between 7 and 21 m·s⁻¹ (Banta et al., 1992), consistent with modeling studies indicating maximum values of about 17 to 21 m·s⁻¹ (Zhang et al., 2019).

In our measurements, differences may stem from our strategy of focusing on fires during their initial stages. We aim to capture the plume top and assess firefighters with the pyroconvective prototype potential rather than focusing on maximum updraft core values in mature plumes. In this approach, we capture plumes in their initial stages, at heights of 1500 to 4000 m, with the plume top located at the ABL top and deepening into the free troposphere. Those plumes lack the extreme cores that are measured with mature plumes (Lareau et al., 2024). Indeed, sondes traveling with the indraft flow into the updraft of a plume, may not reach the central cores. Instead, they may travel in less intense updrafts around the central cores, as illustrated by continuous radar measures (Lareau et al., 2024) and theorized by pyroconvective models (Freitas et al., 2009; Tory and Kepert, 2021).

### 4.3 Sensitivity of updraft profiles to the launching site with respect to the indraft origin

Our research confirmed that the location of the in-plume sondes—whether they are launched at the head, flank, or rear of the fire front—significantly affects the profile of state variables. This effect, previously noted in plume simulations (Canfield et al., 2014; Clark et al., 1996), is evident in both the vertical velocity and temperature differences of the indraft flow relative to the head fire.

We observe distinct temperature spikes in the lower 50% of the head and flank indraft profiles (Figures 5c and 5d). The excess $\theta_v$ difference between the head and rear indraft profiles shows that the head fire generates an updraft, which is quickly replaced by indraft flow from behind. The profile observed at the rear indraft (Figure 5b) reveals an apparent contradiction: although it shows the best- and fastest-rising profile (Figure 6b), it exhibits smaller differences in $\theta_v$ between the ambient and updraft values. This observation suggests that the rear indraft sonde measures the flow of ambient air entering the updraft without a temperature excess during the first half of the profile height. The observation is supported by previous works signaling the rear indraft as the most important and being formed by descending air into the plume neck base (Charland and Clements, 2013; Clark et al., 1996; Clements, 2010; Werth et al., 2011).

These findings highlight the importance of our approach in delivering valuable research data for understanding pyroconvection dynamics.

These temperature spikes (head and flank indraft) and lack of temperature excess (rear indraft) pose challenges for calculating parcel methods and may lead to unrealistic parcel trajectories. Therefore, caution is required when applying the parcel method in the head fire and flank fire indrafts. Our observations indicate that the ML parcel method is the most suitable for raw sonde profile data (Leach and Gibson, 2021; Tory and Kepert, 2021). However, for fireline safety assessment and awareness build-up, head and flank spikes may be considered to analyze worst-case scenarios (Figure 12).

### 4.4 Assessing pyroconvection prototypes transitions

The systematic pairing of ambient and updraft state variables profiles enhances our understanding of fire-atmosphere coupling effects, as well as the application of plume and pyroconvection models. By accurately characterizing local ambient conditions, assessing updraft buoyancy dilution, and determining current plume height through the updraft rising velocity profile , the in-situ collected data significantly improves the traditional fire manager's use of skew-T diagrams and parcel methods in conjunction with the $\theta_v$ profile (Goens and Andrews, Patricia L, 1998; Leach and Gibson, 2021; Tory & Kepert, 2021). We can acquire an awareness of the difference between the current measured plume height, the potential height in the current profile by the parcel method and the height needed for transitioning to a deeper pyroconvection prototype. Additionally, profiles of relative humidity (RH), wind direction, and wind speed contribute to our analysis. They help illustrate how fire-atmosphere interactions are altering the atmosphere boundary layer (ABL) into a fire-dominated ABL Furthermore, we can determine whether the wind is influencing and tilting the plume or if the plume itself is modifying the height of the wind shear level (Figure 12).

Real-time in-situ measurements are crucial for effectively managing the sources of uncertainty that can abruptly affect fire spread and potentially lead to firefighter entrapments (Castellnou et al., 2019). These uncertainties stem from two main factors: changes in the ABL triggered by the fire itself, which can lead to pyroconvection prototype transition (Figures 9 and 10), and changes by atmospheric conditions that are advected into the ABL, influencing these pyroconvection prototype transition (Figures 4 and 12).

The assessment of the pyroconvection prototype, based on plume-top analysis using single-sonde trajectories, has been validated through simultaneous sonde case studies (Figure 11). This validation includes an evaluation of the plume top estimation error (Figure S6 and Table S3). The findings, despite the plume top being a dynamic entity, confirm the reliability

of the pyroconvection analysis, even when factors such as entrainment turbulence at the boundary layer top may affect the readings from the sondes. Importantly, the variability in plume-top measurements aligns with results from alternative methods, such as radar and satellite observations. This observed variability should be considered to effectively address transitions in the pyroconvection prototype.

The different cases analyzed indicated that the pyroconvection transition is highly sensitive to the plume updraft strength (Figures 9 and 10). The updraft strength is closely tied to surface fire activity, defined by front size and depth, as supported by plume model analyses in the literature (Badlan et al., 2021; Rio et al., 2010). The bigger the front feeding the plume, the more protected from detrainment and the less diluted the plume core in its ascend, and because of that, the deeper the plume penetrates into the free troposphere (Liu et al., 2010, 2012).

Our methodology links pyroconvection potential to observed front size, enabling firefighters to evaluate how changes in front size may impact pyroconvection and fire spread. This capability helps to detect a common type of fire resulting in fatalities (Page et al., 2019): small fires that rapidly escalate without changes induced by new advected ambient conditions. They are not restricted by the thermodynamics of the atmospheric vertical profile but rather by the flaming front, which does not provide sufficient updraft strength to travel undiluted to the LCL height and trigger a pyroconvection prototype transition. In such fires,

a change in slope, fuel, or surface weather can intensify the plume updraft strength. This can lead to thermodynamic changes that facilitate a transition to pyroconvection, unexpectedly altering the spread of the fire (Figure 10), and potentially trapping firefighters.

## 4.5 Limitations

The safety requirements and challenges of navigating through a rapidly spreading fire landscape conditioned the use of in-

plume sondes information. Data collected, primarily during the early stages of the fire, has been used in 87% of cases to raise awareness and adapt tactics. In contrast, 13% of the cases utilized this data for last-minute safety alerts (Table 4).

Gathering data directly within a fire environment poses challenges to the safety of the launch team and the reliability of the collected data. Our method has four potential limitations.

First, verifying whether the data obtained from the sonde is adequate for use as an in-plume or environmental profile is

important. These sondes must ascend an average of 300 meters (Figure 6) to properly position them in the updraft. This limitation directly affects our ability to assess the plume's height if it does not rise above the minimum required level or the plume is too weak to separate its rising velocity values from ambient ones easily.

Additionally, well-established large fires can have multiple updrafts (Krishna et al., 2023) leading to pyroconvection profiles. With our method, we can only assess the updraft over the plume that our sonde has ascended.

Moreover, it is crucial to account for temperature spikes in the sonde's initial ascent path, particularly in flank and head indraft profiles when using this data for modeling; otherwise, numerical computations of Rib and fireCAPE may be inaccurate.

Launching multiple sondes can address all three limitations (Figure 11).

Lastly, during existing extreme pyroCb events, safety during launch may be compromised by the extremely unpredictable behavior. While we still have the capability to launch reliable in-plume sondes, this is limited to rear-indraft sondes (Figure

S9).

## 5 Conclusions

We present a new observational method and strategy aimed at enhancing awareness of pyroconvection and improving understanding of fire plume dynamics and their interactions with the surrounding environment. The method is based on simultaneous sounding observational profiles of in-plume wildfires and their surrounding ambient. These profile observations

enable us to complete a description of the main dynamic characteristics of the fire plume relative to the ABL characteristics and to classify fires according to pyroconvection prototype categories.

Despite the limitations of sondes as a single trajectory inside the plume, the results from 173 successful sondes offer robust evidence for reliably detecting plume top heights using the sonde rising velocity, wind, potential temperature, and humidity profiles.

Compared to previous radiosonde applications in areas affected by fires, the novelty of this approach lies in the systematic and simultaneous collection of data from ambient conditions and updraft profiles within the plume. By employing this dual-sounding method, we gather observations of the fire-atmosphere dynamic interactions in almost real time. This coupling is missing in atmospheric models. More specifically, our observations and analysis enable us to quantify the rapid vertical variations in moisture and wind profiles driven by land-sea contrasts, topography, frontal advection, and their interaction with

the fire. This in-situ quantification is crucial for assessing potential transitions to deeper convection, which may drive extreme fire behavior.

Our new in-plume radiosonde methodology for profiling state variables provides a cost-effective and essential complement to current assessment methods during wildfire operations. It enhances the understanding of fire-atmosphere dynamics in-situ and in-real time, thereby reducing uncertainty and increasing safety for firefighters confronting increasingly intense wildfire events

worldwide.

### Data availability

Final Dataset in EWED project data portal: http://wildfiredataportal.eu/
The profiles in the Figures are in DOI **10.5281/zenodo.17886250**

**Funding**

This project has received funding from Directorate-General for European Civil Protection and Humanitarian Aid Operations (ECHO) through the project EWED (101140363), and the Horizon 2020 research and innovation program under grant agreements No 101037419 (FIRE-RES).

### Conflicts of interest

The authors declare that they have no conflict of interest.

### Authors contribution

MC, MM, MB, BR planned the campaign; MC, MB, BR, JP, LE, BV, and PG conceptualized the methodology and performed

the measurements; MC, MB, PG, and JV analyzed the data; MC wrote the manuscript draft; JV, MJ, CVH, TR, CS, MB, PG, BB, and JP reviewed and edited the manuscript.

### Acknowledgments

The authors thank the Cos de Bombers de la Generalitat de Catalunya, particularly the GRAF units, for their support during research and testing. We also acknowledge the Servei Meteorològic de Catalunya for radar data access. Special thanks to

Antonio Ariza and Alex Sancho for adapting radiosonde equipment for the fire service, to Ruth Domènech Jardí for reviewing data management, and to Jorge Saavedra and Jonatan Troncho for facilitating data collection during Chilean wildfire

campaigns. We also appreciate Jose Cespedes for providing helium and equipment, and Vicor Lopez for capturing the launching procedure.

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
