# Peer review of "Integrating Fireline Observations to Characterize Fire Plumes During Pyroconvective Extreme Wildfire Events: Implications for Firefighter Safety and Plume Modeling"

_EGUsphere, 2025_

## Author Comment (AC1)

**Overview**

This preprint presents a novel and practical methodology for characterizing pyroconvective wildfire plume dynamics using dual radiosonde soundings (in-plume and ambient). The study spans 156 field launches across four countries between 2021 and 2025 and offers both operational and scientific insights into plume development, real-time hazard awareness, and fire-atmosphere interaction modelling.

The manuscript is timely, rigorously detailed, and bridges a rare and valuable gap between operational field constraints and mesoscale meteorology. The work is distinguished by its applied innovation, extensive empirical validation, and potential to substantially inform firefighter safety procedures.

**Strengths**

1. Novel Methodology with Operational Value

- The use of paired in-plume and ambient radiosonde profiles is both innovative and cost-effective, rendering it feasible for deployment during active wildfires.

- The operational integration into tactical decision-making workflows sets this study apart from traditional simulation-based or laboratory-bound research on pyroconvection.

2. Robust Field Campaign

- With 156 launches covering a diverse range of vegetation types, meteorological conditions, and terrain profiles, the dataset represents an impressive empirical foundation.

- The inclusion of both prescribed burns and uncontrolled wildfires increases the method's general applicability.

3. Validation Through Multi-Modal Comparison

- Plume-top altitudes inferred from vertical velocity profiles were validated against radar echotop data, which significantly strengthens confidence in the method.

- Application of parcel theory for forecasting potential plume development (e.g., pyroCu onset) is methodologically sound and well-executed.

4. Classification Framework

- The six-category plume prototype typology (based on ABL height, LCL height, and wind shear layers) is operationally intuitive and scientifically coherent.

5. Actionable Outcomes

- Several case studies (e.g., Martorell and Santa Ana) demonstrate that tactical decisions informed by the sonde data likely contributed to risk mitigation. This real-world applicability is a major strength.

*We thank the reviewer for the summary and very positive assessment of our work, and specifically the recognition of our methodology and data set.*

**Weaknesses / Areas for Improvement**

1. Clarification of Balloon Typology (Ref. Line 145, Table 1)

- The classification "professional high-altitude balloon" is misleading. A more accurate term would be **"operational radiosonde systems"**, as both small and large balloons may be used professionally. For instance, an MW51 Vaisala ground station combined with an RS41 sonde and a Totex TA50 or TA100 balloon could easily reach 400 hPa level, albeit with higher helium demand.

   *Thank you for your input. We aim to clarify the misleading terms by using the proposed "operational radiosonde systems." In Table 1, instead of 'Professional high-altitude balloons'*

   *Although we are aware of the various types of radiosondes that are referred to by the reviewer, our reasons for designing and developing the light system are related to the safety requirements set by the aerial controller of the fire.*

   *In line 136, where it said: 'Safe for operating along with aerial resources'*

   *We will add the next new text:*

   *"Ensure compliance with specific safety requirements that may differ from general aerial control regulations. These are proposed by the fire service aerial coordination for operating alongside firefighting aerial resources: radiosondes weighing less than 50 grams and colored balloons with a capacity of less than 90 liters. Note that these requirements may vary internationally, and we adhere to the strictest standards."*

- Table 1 also incorrectly claims that such systems are incapable of simultaneous launches. In fact, MW51 systems can track up to four sondes concurrently and have portable variants.

   *We will adapt the table accordingly.*

- Furthermore, the claim that professional radio-sounding systems are "not safe for aerial resources" is inaccurate. All operational weather balloons (regardless of size) comply with aviation safety regulations. In contrast, marking helicopters as inherently "safe" for aerial operations is misleading, as such platforms require **strict coordination with air traffic control and firefighting aviation assets**. These inaccuracies should be revised to reflect standard aviation safety protocols.

   *The aerial control during a wildfire establishes the specific requirements for the simultaneous operation of drones and radiosondes alongside helicopters and planes. As stated in the previous clarification in the first paragraph of this section, the radiosonde system must meet the safety requirements specified by the aerial coordination within the fire service*

*team. This adaptation involves using the lightest possible radiosondes and colored balloons to ensure visibility.*

2. Reliability of Windsonde Data During Descent (Ref. Line 190)

- The authors should clearly state that **measurements during descent are generally considered less reliable**, even for professional sondes. Windsonde systems have not been formally validated for descent-phase data collection.

*In the new modified text, we acknowledge that such measurements are less reliable and can exhibit discrepancies of tens of meters. However, in our study, we have achieved a satisfactory accuracy within the profiling. This has allowed us to propose and apply a classification of pyroconvection prototypes (Castellnou et al., 2022), which is the primary goal of the proposed methodology. Although less reliable, profile measurements taken during descent still enable us to identify key metrics in the fire-weather interaction, such as the ABL and LCL heights (with 82 m uncertainties in their height estimations).*

*Please note that our methodology includes launching separate radiosonde not influenced by the fire conditions to obtain complete and more comprehensive observational evidence of the interaction between the plume and the surrounding environment. However, if due to whatever circumstances this is not successful or possible, we can use the measurements taken during a sonde's descent (attempting to endure those outside the plume) as a best approximation. Such measurements are a much better estimation than ambient radiosondes from 10s to 100s of km and hours of difference, as normally used as reference in pyroCu studies (Lareau and Clements, 2016; Tory et al., 2018)*

*We provide additional complementary materials, statistically comparing the profiles of ascent and descent of the same sonde in Figure 5 as detailed below.*

*Accordingly, we change the text in line 190:*

*Old version:*

*'Ambient sonde:*

*Launched outside the fire influence (Figure 2), it measures the vertical profile of the state variables in an environment uninfluenced by the fire plume.*

*Although launching a separate ambient sonde is recommended, our campaign findings indicate that an ambient profile can also be obtained from the in-plume sonde descent path if the sonde is cut-down once it is outside the plume's influence. By comparing data from both the in-plume descent and the ambient sondes, we can improve the reliability of our findings.'*

*New version:*

*'Ambient sonde:*

*Launched outside the fire influence (Figure 2), it measures the vertical profile of the state variables in an environment uninfluenced by the fire plume. By comparing data from both the in-plume descent and the ambient sondes, we can improve the reliability of our findings.*

*Although launching a separate ambient sonde is recommended, our campaign findings suggest that it may sometimes be operationally impractical. However, an ambient profile can also be obtained from the in-plume sonde descent path if the sonde is cut-down once it is outside the plume's influence. Although less reliable (mean absolute error: 82 m), analysis of such profiles measurements taken during descent still enables us to identify key metrics in the fire-weather interaction, with acceptable variable uncertainty (Figure S5)*

*Proposed Figure S5:*

[Figure]

| variable | Mean of differences | Std deviation of differences |
|---|---|---|
| Ө (K) | 0.78 | 0.63 |
| RH (%) | 2.22 | 6.35 |
| q (g·kg-1) | 0.07 | 0.68 |
| WS (m·s-1) | 1.22 | 2.21 |

*Figure S5: Validation of descending ambient sonde profiles. Evaluation of the mean and the standard deviation between appropriate (ascending) ambient sondes and descending ambient sondes collected during the same fires. The results indicate that the measurements are acceptable for the assessment of pyroconvection prototypes. Such profiles are essential when ambient sonde data for comparison with in-plume profiles is unavailable. In these instances, in-plume sondes descending outside the fire area can supply needed ambient data.*

- As shown in Bessardon et al. (2016, Kumasi campaign), **ground-based reference measurements** of pressure and temperature were used to calibrate Windsonde outputs. It is unclear whether a similar calibration procedure was applied in this study.

> *The use of Windsonde was tested and calibrated against Vaisala RS41 sonde used by MeteoFrance team during the 2021 LIAISE campaign (Boone, 2019) of ABL measurements in Lleida (Spain). The windsonde system was adjusted in the field and the measurements showed reliable results. (Castellnou et al., 2022)*

- Moreover, the same research highlights concerns regarding **wind speed and direction accuracy**, particularly during turbulent or shear-laden environments. The authors should explicitly discuss whether and how these limitations were addressed.

  > *We acknowledge in the text that turbulence in fire-weather conditions can lead to noisy soundings. Bessardon et al. (2016) highlighted this issue, particularly with irregular patterns in measured horizontal wind speed. They recommend using smoothed lines for analyzing wind speed and direction, as we apply in our analysis. These two parameters are relevant but not crucial for identifying pyroconvective prototypes and have not hindered the analysis of Windsond data during wildfires.*

- Known **instrumental limitations** of Windsonde include weak response to rapid humidity and temperature changes, and **systematic underestimation of altitude (up to ~40 m)**. These should be acknowledged and addressed to justify continued use of this platform.

  > *The instrumental limitations for rapid humidity and temperature changes, as well as understimations of altitude, have been well-tested and identified during the LIAISE campaign and in the 2021 fire when comparing radiosonde agreement.*

  > *The systematic underestimation of 40 m is not significant for the operational use in identifying the pyroconvection prototypes.*

  > *The weaker response to rapidly changing temperature and/or specific humidity with Windsond is not significant for operational use of the sonde (Figure S5), which show well-defined in-plume and outside-plume data.*

  > *While the Windsond is known to exhibit a slower response when moving from a cloud to a warmer, drier environment (Bessardon et al., 2019), our study found that the height of this transition effectively serves our purpose. Furthermore, pyrocloud tops identified by radiosonde measurements aligned well with radar data (Figure 8) with a mean absolute error (MAE) of 166,7 m, validating the Windsond's capacity to provide the operational information we need.*

  > *Following the discussion in the last three bullets in this section, we will update the original text between lines 153 and 156:*

  > *Old version:*

  > *'The system has been previously tested against larger, professional radiosondes and successfully achieved relevant measurements, despite its*

*weaknesses in GPS processing and humidity response time at cloud tops (Bessardon et al., 2019). Previous research that we conducted during active wildfire events demonstrated that these challenges did not hinder the detection of pyroconvective phenomena (Castellnou et al., 2022).'*

*New version:*

*'The instrumental capabilities of the system have been previously tested against larger radiosonde systems, as RS41, during the LIAISE campaign (Boone et al, 2019) of ABL measurements in Lleida (Spain). Results showed a strong profile agreement between both radiosondes systems (Castellnou et al. 2022). While certain weaknesses, such as a 40-meter altitude underestimation, issues with GPS processing, slow humidity response at cloud tops, and noisy wind profiles in turbulent conditions (Bessardon et al., 2019), were noted in the plume turbulent conditions, they were not detrimental to the accuracy of identifying pyroconvective prototypes during wildfires (Castellnou et al., 2022).*

*To continuously validate the Windsond operational effectiveness, we systematically record plume measurements using fire service planes and radars whenever possible*

3. Quantitative Predictive Success Rate

The manuscript would benefit from a summary table or appendix quantifying:

- How many launches successfully entered the plume core?

- How many failed or produced partial profiles?

- In what proportion of cases did fire development escalate to **Extreme Wildfire Events (EWEs)**?

- In how many instances did the radiosonde data lead to a tactical change (e.g., crew withdrawal)?

   *Due to the length of the table with 156 sondes, we will add to the dataset repository of the paper a detailed table for the sondes launched using the next headings-:*

   ***Fire:*** *name of the fire and type: WF (wildfire), PF (prescribed fire). If the wildfire incorporates the (EWE) means it evolved to an extreme wildfire (EWE) of category IV or higher (Tedim et al., 2018)*

   ***Lat & Long:*** *latitude and longitude coordinates*

   ***Data and hour:*** *identifies launching day and hour*

   ***Sonde type:*** *identifies in-plume sondes types (rear-indraft, flank indraft and head indraft), ambient sondes and umbrella sondes as described in Figure 2. Due to this specification, Figure 2 and its description is modified to include the umbrella sondes.*

   ***Success:***

*S: success, sonde entered the plume.*

*N: NO success, sonde didn't when into the plume. If sonde failed reason is provided.*

**Motivation to launch**: *goal of the launching*

**Impact on decision:** *use of the information provided by the sonde. We classify it as:*

*Awareness, -sonde provides awareness of the real ambient conditions on-site and real plume deepening into the ABL or FT.*

*Tactical, sonde information forces tactical adjustment in operations to avoid the impact of potential pyroconvection changes in fire behavior.*

*Safety, when sonde is used to confirm ongoing pyroconvection transition*

*Example of the proposed table (the table in the repository totals 166 sondes):*

| fire | Date and hour | Lat & long | Sonde type | Success | Motivation to launch | Impact on decision |
| --- | --- | --- | --- | --- | --- | --- |
| | | | | | | Reason to failure |
| WF (EWE) Martorell (CAT) | 13/07/2021 16:09:22 | 41.465158 1.936343 | In-plume Rear indraft | S | Validate pyroCu deepening | Tactical |
| WF (EWE) Martorell (CAT) | 13/07/2021 16:25:11 | 41.453402 1.943707 | Ambient | S | | |
| WF (EWE) Martorell (CAT) | 13/07/2021 17:05:07 | 41.467680 1.921600 | In-plume Head indraft | S | Validate potential transition to pyroCu | Tactical Safety |
| WF Torroella de Montgrí (CAT) | 22/07/2021 18:42:03 | 42.071343 3.145890 | In-plume Head indraft | S | Validate potential transition to pyroCu | Awareness |
| WF (EWE) Sta Coloma de Queralt (CAT) | 24/07/2021 19:17:38 | 41.529135 1.383498 | Ambient | S | | |
| WF (EWE) Sta Coloma de Queralt (CAT) | 24/07/2021 19:41:55 | 41.528420 1.453758 | In-plume Flank indraft | S | Validate potential transition to pyroCu | Awareness |
| WF (EWE) | 25/07/2021 10:40:47 | 41.510465 1.486245 | In-plume Rear indraft | S | Validate potential transition to pyroCu | Tactical |

| | | | | | | |
|---|---|---|---|---|---|---|
| Sta Coloma de Queralt (CAT) | | | | | | |
| WF (EWE) Sta Coloma de Queralt (CAT) | 25/07/2021 18:19:59 | 41.517753 1.494428 | In-plume Rear indraft | S | Validate pyroCb transition and strenght | Safety |
| WF La Pobla de Massaluca (CAT) | 12/08/2021 15:10:09 | 41.224933 0.332295 | In-plume head indraft | N | Validate pyroconvection prototype | Awareness Launch into a weak-intermitent indraft |

*A summary table with statistics about success and use will be added in the results section 3.4. Such section is changed accordingly to 'Usability of plume profiling methodology':*

*Old text:*

*'3.4 Failed profiles*

*It is important to note that during the campaigns, we did not observe detrained sondes from the plume once the sonde entered the plume neck. However, we have had cases of sondes failing to enter the plume or entering the plume at higher altitudes when we launch into weak or intermittent indraft conditions. Those cases have always been reported with launching conditions too far away from the head fire (Figure S4) or when we launch into a decaying head fire, and there are strong surface winds present (>6 m·s-1). '*

*New text:*

*'3.4 Usability of plume profiling methodology*

*Over the four years of fire campaigns during which we tested our methodology, we obtained clear results supporting the use of paired ambient-in-plume profiling with radiosondes on active wildfires (see Table 4). The low failure rate of 7.73% and the consistent application of sonde information for awareness and safety indicate that this methodology is well-suited for adapting operational tactics—utilized in 39.7% of our case studies—to address the challenges posed by pyroconvection transitions.*

*It's important to note that during the campaigns, sondes that failed to enter the plume did so due to being launched too far from the plume base into weak or intermittent indrafts (Figure S4.) and normally in the head or flank indraft. Rear indraft sondes, that better capture the main indraft into the plume can endure longer distances.*

*Table 4.- Summary of success and use in decision making of the sondes launched (to be completed).*

| Type of sonde | Proportion over total sondes | description | |
|---|---|---|---|
| Failed sondes | 7.73% | 61.3% too weak indraft, or launching too far away | |
| | | 23% pushed to the ground by rear indraft | |
| | | 15.3% due to sonde failure | |
| operational | 73.27% | Awareness | 34.1% |
| | | Tactical | 32.7% |
| | | Safety | 7% |
| Research | 19% | | |

4. Reproducibility: Launch Schedule and Decision Criteria

- For reproducibility and model intercomparison, the authors should provide a **complete launch schedule overview**, including exact timestamps and GPS coordinates of each sonde release.

- Additionally, the **criteria used to determine the moment and location of launch** should be explicitly stated (e.g., wind indicators, visual cues, forecast thresholds).

- These criteria appear to be field-operational in nature, but formalising them would help transfer the method to other contexts.

   *The launch schedule data is presented in the table mentioned in point 3. The criteria for launching the sondes, outlined in the 'motivation to launch' field of the table, are based on the operational need to validate the pyroconvection prototype's potential and the likelihood of transitions.*

5. Real-Time Workflow and Decision Chain

The article does not describe the **complete operational workflow** from launch to decision. Clarifying the following would significantly improve transparency:

- How is data transmitted to and from the operations centre?

- Who is expected to perform the data analysis (on-site, centralised, or remote team)?

- What additional data sources are used (e.g., satellite, radar, fireline reports)?

- What is the **end-to-end latency** between launch and actionable tactical insight?

- Are any **supporting information systems or software platforms** (e.g., for visualisation or alerting) required or recommended?

   *In the methodology section, we have added the following information:*

   *'The sonde operational workflow includes the fire analyst being part of the launch team, allowing immediate analysis of observational data collected during the sounding. If the analyst is not present, data is uploaded to cloud storage from field mobile devices for command post analysis. The analyst reviews the vertical profiles to approve or adjust ongoing operations in collaboration with the incident commander and safety officer. Additional*

*information is gathered from fireline crews, drones, planes, and meteorological radars, when available. Data management should occur within one hour of the in-plume launch, with a two-hour reference limit. The process involves data transfer, visualization software for profiles, and cloud archiving to make the observations accessible to the incident management team.'*

6. Sonde Sampling Bias

- The authors acknowledge that sondes may not always enter the plume core, which may skew thermal and vertical velocity readings. Further statistical quantification of this sampling uncertainty would be beneficial.

  *We acknowledge in the discussion that sondes may not always enter the cores of the plumes. As a result, the readings obtained may underestimate the thermodynamic and vertical velocity characteristics of the more buoyant core inside the fire plume. However, the vertical velocity profile still accurately indicates the plume top height, with discrepancies of only a few hundred meters, which falls within the typical variability of plume tops.*

  *In the manuscript, we detail the launch procedure for accurately measuring the fire plume. As discussed in the modified results section 3.4, the key to success is ensuring the sonde penetrates a well-established plume indraft. This typically requires proximity to the plume (see the added sondes table in the dataset for failure reasons).*

  *It is important to notice that the indraft requirement can't be defined numerically and launching can only proceed when the indraft is physically experienced in the launching site. Sondes launched during the Guisona wildfire on July 1, 2025 into a strong 18 m·s-1 indraft wind, faced significant horizontal trajectories of 9 kilometers within the indraft before entering a plume that was measured having a top at approximately 11200 m AGL by radar. Conversely, sondes released outside or in intermittent indraft from plumes with shallow plume tops (1,000-2,000 m) often failed to reach the plume, needing launching position closer to the plume neck (up to 300 m) to succeed.*

  *This information is added to section 3.4 and the modified complementary figure S4*

7. Terminology

- Some terminology (e.g., "$\theta v$ spike", "fireABL", "S parcel") may not be immediately clear to the broader meteorological or fire-behaviour audience. A glossary or summary table of variables and acronyms is recommended.

  *Thanks for the observation. We will complement Table 2 and include descriptions so it becomes a complete table of variables and observations describing terms and providing symbols and units to facilitate the reading*

[revised manuscript text omitted]

**Suggested Revisions**

- Replace ambiguous or inaccurate entries in **Table 1**, particularly regarding balloon classifications, safety, and simultaneous sounding capability.

    *Done in point 1*

- Add a **summary table** of all launches with fire name, time, coordinates, outcome, and tactical decision if applicable.

    *Done in point 3*

- Include **error metrics** for vertical velocity-derived plume tops compared to radar.

    *We provide the data in the new updated Figure 8:*

[Figure]

Correlation between plume top measured height and radar measured plume top

- Clarify how and when real-time analysis was conducted, by whom, and how long it took.

  *See Table in point 3*

- Address the known **technical limitations** of Windsondes and justify their use despite weaknesses.

  *Done in point 2*

- Provide access to a **full launch log and reproducibility protocol**, including selection criteria for launch timing and location.

  *Done in table in point 3*

---

## Author Comment (AC2)

I would like to congratulate to the authors for this work that contributes with data analsysis for plume analsysis in real time in real scenarios for decision-making support.

The article proposes a sounding methodology to characterize fire plumes during wildfire events. Although it does not provide a detailed protocol, it clearly emphasizes the importance of in-situ data collection using a feasible and affordable approach to better understand and assess plume behavior and atmospheric dynamics during a fire. I find this work highly valuable, as fire analyses are often based solely on forecast data. This study offers a practical method for estimating fire plume dynamics in areas where radar systems are prohibitively expensive. Additionally, the real-time data collection and analysis performed during the fire enhance its value for emergency management, moving beyond the traditional post-event analysis approach.

The article presents very useful information and includes clear application examples. It is well-written and well-structured. However, readers should be familiar with fire and atmospheric terminology, and ideally be up to date with the current state of research on fire plume dynamics and fire typologies.

The article outlines a method for obtaining atmospheric soundings from an ongoing fire, including measurements not only of the ambient atmosphere but also of the head, flank, and rear of the fire. While the work does not go into detail about the balloon sounding launch procedure or the success rate—points also raised by an anonymous reviewer-it focuses on the sounding data collected from different locations and their use for analyzing plume state and potential, which is useful for operational decision-making.

The authors acknowledge the limitations of their approach and provide recommendations for how to work with the data presented.

This article makes an important contribution to operational practices and this research field by characterizing fire plumes and assess transitions between different fire plume behaviors.

I would recommend to the authors to apply at least minor comments when they apply.

*We would like to thank the reviewer for taking the time to review the paper and for providing valuable comments that have helped us clarify and improve the proposed methodology and its explanation.*

The article focuses on integrating sounding data to characterize fire plumes, as stated in the title, and the content aligns well with this aim. The authors address the inherent uncertainty in the sounding data, particularly due to the balloons moving within the plume's indraft and temperature peaks, which affects the success rate of non-ambient soundings. However, there is no quantitative assessment of the uncertainty for each variable derived from the sounding methodology, nor an analysis of how sensitive the plume top estimation is to those uncertainties—beyond potential temperature and rising speed. I understand this could be the focus of a separate study, which would require a large number of soundings and additional instruments for validation in different contexts with prescibed fires. Still, the article highlights the importance of potential temperature and vertical velocity as key factors in plume-top estimation, with other variables being more relevant when using the parcel method for potential plume-top prediction.

*In the revised version, as commented on with the first anonymous review, we have added the following information.*

*1.-Quantification of the Uncertainty of plume top height. The uncertainty has been estimated using the radar data. It shows that there is an average error of 166.8 m*

*2. Quantifying the uncertainty of the variables measured under ambient conditions by the descending sondes has been achieved by comparing ascending and descending data. This comparison justifies the use of the descending profile as ambient data, if necessary, to complement the characterization of the thermodynamic profiles.*

*In section 4.2, we have already discussed the fact that sondes represent a single trajectory inside the plume. Compared with radar readings, the sondes may not penetrate the most active buoyant cores. However, they are definitely transported within the plume. Therefore, they are able to detect and estimate the plume top (compared with radar) and the average rising speed.*

*To improve the assessment of the uncertainty of the sondes and the trajectory, radar measures will be required to obtain a continuum of the plume. As stated by the reviewer, such a work will be completely new research, not in the scope of this paper. We have a plan to do it in the future.*

*In the current manuscript, we have focused on providing a methodology to characterize in situ the plume thermodynamic variables. In doing so, we are able to determine the pyroconvection transition potential using easy-to-operate sondes in the fire environment without the need to deploy in the field for special teams. We demonstrate that such measures do provide an increase in awareness and capacity to provide accurate information for decision-making.*

*Despite the limitations on uncertainties in the plume top and lack of capacity to capture the highest vertical velocities at the plume core, our methodology demonstrates that by accounting for the indraft where the sonde is launched, we obtain data accurate enough to capture the plume top. This information is added to a classification of the pyroconvection prototype and its potential transition during the fire operations. Moreover, it has now been used to constrain and evaluate the results of fire plume models to advance our understanding of the interactions between environmental conditions and the rise of the fire plume.*

*To include this information on the uncertainty of our observations gathered by the sounding system, we will complement this in the revised version of the paper with supplementary material. In this SI, we analyze the observations of sondes launched simultaneously. In studying several collocate and simultaneous soundings, we are able to compare the accuracy of different trajectories and the robustness of our thermodynamic profiles.*

**New Complementary material:**

**S7.-Uncertainty assessment for the radiosonding system**

*We assessed the uncertainty in our radiosonde data by analyzing measurements taken from radiosondes launched simultaneously from the same location. Our analysis included five ambient soundings and three in-plume sondes. We calculated the mean and standard deviation of these measurements to better understand the potential variations during the ascent of the sondes. This approach allows us to quantify deviations that could impact our results.*

*For all the variables calculated, the uncertainty observed is low (Table S7.1). The large uncertainties are primarily found at the top of the atmospheric boundary layer (ABL). Here*

we find variations that range from 200 to 300 meters (Figures S7.2 and S7.3). This level of uncertainty is typical, as both the ABL and plume top are not constant and tend to fluctuate in the entrainment zone, which can be characterized with air masses with three different thermodynamic characteristics: fire plume, ABL ambient, and free tropospheric air. The uncertainty analysis enables us to contextualize the information obtained, suggesting a buffer of 200-300 meters in the estimation of the plume top when assessing the potential for pyroconvection transitions.

**Table S7.1**. Uncertainty of radiosonde trajectory for ambient and in-plume measures for the variables used in the radiosounding methodology. The variables acronyms are Tpv as virtual potential temperature, RH as relative humidity, WS as wind speed, WD as wind direction and in the case of in-plume sondes we add vertical velocity. Ambient uncertainty has been obtained by ambient sondes on the 9th-08-2025, launching five consecutive sondes between 16:03 and 16:11 UTC, at 46º 03 45.44'' N and 0º 40' 22.54''E (Spain). The in-plume sondes are obtained for three sondes launched during the Casablanca fire the 08-02-2023 in Chile (See Figure 11) between 21:46 and 21:51 UTC from the same spot.

| | T (C) σ | RH (%) σ | WS (m·s⁻¹) σ | WD (º) σ | Vertical velocity (m·s⁻¹) σ |
|---|---|---|---|---|---|
| Ambient | 0.204 | 0.818 | 1.08 | 12.16 | |
| In-plume | 0.379 | 0.425 | 1.119 | 20.924 | 0.416 |

*1.- Uncertainty in measuring ambient conditions*

[Figure]

**Figure S7.1**.- Profiles of the main variables that characterize the ambient conditions: virtual potential temperature (Tpv), relative humidity (RH), wind speed (WS), and wind direction (WD). The profile shows the mean ensemble average based on five soundings (and the standard deviations). The five soundings were launched between 16:03 and 16:11 UTC. The uncertainty is quantified by the standard deviation of the mean at every 50 m of altitude.

*2.- Uncertainty in measuring in-plume conditions*

[Figure]

*Figure S7.2.- Profiles of the main variables that characterize the in-plume conditions: virtual potential temperature (Tpv), relative humidity (RH), wind speed (WS), and wind direction (WD), and vertical velocity (w). The profile shows the mean ensemble average based on five soundings (and the standard deviations). The three soundings were launched between 21:46 and 21:51 UTC in the Casablanca fire (Chile) (see Figure 11). The uncertainty is quantified by the standard deviation of the mean at every 50 m of altitude.*

Minor Comments:

Update model name and cite missing reference: MESO-NH/Forefire – Jean-Baptiste Filippi.

*Thanks for detecting it. Done. We added the next cite:*

*Filippi, J. B., Mari, C., & Bosseur, F. (2013, July). Multi-scale simulation of a very large fire incident. Computation from the combustion to the atmospheric meso-scale. In 4th Fire Behavior and Fuels Conference.*

There is no need to justify the balloon radiosounding method beyond its affordability and operational simplicity. Other technologies like high-altitude atmospheric balloons, Doppler radar, UAVs, and helicopter-mounted sensors are capable of real-time data collection as well.

Table 1 may be unnecessary, but I understand that in real-world field campaigns, time constraints, costs, and complexity support the value of the information presented.

*We acknowledge that other methods can collect data in real time. However, they have logistical challenges, as highlighted in Table 1, or fail to accurately identify the plume top, such as with helicopter-mounted sensors or UAVs. Among the options, radar and balloons are the most effective systems. Additionally, small balloons are the only ones permitted to operate in a fire environment.*

*We believe that these issues warrant discussion, as in some campaigns, the use of UAVs or balloons has been prohibited during the daytime due to aerial firefighting operations. While radars are a comprehensive solution for data gathering, they are expensive, challenging for*

*regular firefighters to operate, and difficult to transport and install near a rapidly advancing fire front, which presents safety challenges.*

*Considering everything, we believe it is important to explain why small balloon soundings are preferred over other technologies. This is a fundamental aspect that justifies our methodology.*

It would be helpful to use a consistent temperature unit throughout the article.

"S parcel increase by3°C" → should be "S parcel increases by 3°C" (missing space and verb correction).

*We addressed the issue.*

"estimating the dilution plume height, is adequate" → remove the comma: "estimating the dilution plume height is adequate."

*We addressed the issue.*

In Section 3.3.3, regarding the Casablanca III fire in Chile: it states the fire grew to 12,073 ha between February 8–10, 2023, but also says it had already blown up on February 2, reaching 363,000 ha. Including fire spread metrics would help readers better understand the relationship between plume dynamics and fire spread behavior.

*Sorry, the text was not clear enough. We referred to that after the chaotic situation on the 2nd when fires burn 362.000 ha, then between the 8th and the 10th the Casablanca fire grew up to 12703 ha. We intended to show how, amid extreme and huge extreme wildfire conditions, the proposed methodology can still work properly.*

*We addressed the issue .*

***Old text:***

*'A set of updrafts and ambient pairs of sondes were launched at the Casablanca III fire in Chile (Figures 11 and 12). The fire grew up to 12073 ha between the 8th and 10th of February 2023. The situation in Chile was dramatic after the fires blew-up on February 2, resulting in over 362.000 ha burned'.*

***New text:***

*'A series of updrafts and ambient pairs of sondes were launched at the Casablanca III fire in Chile (see Figures 11 and 12). The situation in Chile became dramatic after the fires intensified on February 2, resulting in over 362,000 hectares burned. Additionally, a new fire grew to 12073 hectares between February 8 and February 10, 2023'.*

"Fire behavior was initially expected to calm in the early evening, but there was a 60% chance of intensification during the day-to-night transition due to the advection of drier air from the SW" — I assume this is for context, and the 60% probability comes from a weather forecast ensemble.

*Thanks for highlighting this not enough clarified issue. The affirmation is based on the forecasted chances of the event for that day. That forecast is managed by the planning section in the Incident Command system and passed down to the units.*

***New text:***

*'Fire behavior was initially expected to become less active in the early evening. However, the combined assessment of various weather forecasts indicated a 60% chance of intensification during the transition from day to night. This potential increase in fire activity is due to the movement of drier air from the southwest into the area.*

---

## Author Comment (AC3)

This paper presents a valuable contribution to wildfire science by integrating fireline observations to enhance the understanding of fire plume behavior during pyroconvective extreme wildfire events. The authors effectively highlight the importance of detailed, real-time observational data in characterizing complex fire-induced phenomena, which has significant implications for improving plume modeling accuracy. The study's focus on firefighter safety underscores the practical relevance of accurately predicting plume dynamics to inform operational decision-making. Overall, the work advances both the scientific understanding of extreme wildfire behavior and its application to emergency response and safety management, offering meaningful insights for researchers, practitioners, and policymakers involved in wildfire mitigation and response efforts.

Having atmospheric profile data in the environment and inside the convective plume is of undeniable value, especially since it is emphasized that the indices proposed in recent years to classify pyroconvective activity do not seem entirely satisfactory. It is also important to highlight the significant contribution of integrating numerical weather prediction model data, as these can help to underscore the models' own limitations. Would it be interesting to incorporate atmospheric profiles from models with higher vertical and/or horizontal resolution, such as AROME? On the other hand, meteorological radar measurements also hold great relevance, as they allow for validation the estimates of plume tops. In summary, this is a substantial work that sheds light on research in such a critical field, given the key role that pyroconvective activity plays in the escalation of megafires.

*Thanks for the comments.*

*We evaluated the uncertainty of various numerical weather prediction models. In our analysis, we use the ICON model (13 km spatial horizontal resolution) because it offers (1) improved resolution compared to the GFS model (28 km spatial resolution), (2) better representation of the topography and land use, and (3) can partially resolve deep convection. In addition, it remains a freely accessible global model available to all fire and rescue services. This choice was significant, as we had already applied our methodology to analyze fires in Europe and Chile.*

*We also tested the AROME model, which features a higher spatial resolution of 2.5 km with the advantages mentioned above, resulting in reduced uncertainty in our predictions of the fire-weather interactions.*

*The increase in horizontal resolution among the GFS, ICON, and AROME models results in a reduction of uncertainty (Figure S8) in the vertical profile (75$^{th}$ percentile). The GFS model is underestimating the potential temperature in the lower part of the Atmospheric Boundary Layer (ABL) by 3 K, while ICON underestimates it by 2 K, and AROME underestimates it by an average of 1 K. The uncertainty for RH decreases from 30% for the GFS model, to 20% for the ICON model, and finally to 10% for the AROME model.*

*Evaluating numerical model uncertainties concerning in-situ ambient measurements supports our method of using paired ambient and in-plume radiosoundings to analyze pyroconvection during fires. This provides essential information for safety and strategies during wildfires.*

*Following the advice of the reviewer, and to show the comparison of the in-situ observations with numerical models at different resolutions, we will add a new Figure in the supplementary materials section to address this review concern:*

[Figure]

**Figure S8.- Evaluation of the representativeness of numerical models compared with in-situ soundings of potential temperature and relative humidity**. The assessment is based on the bias of the vertical profiles for both potential temperature and relative humidity. This bias is calculated by comparing the observed profiles with the predictions from numerical models. To quantify the uncertainties, the distribution of differences is analyzed and represented in the plot as mean values and percentiles of the differences between the sonde measurements and the model's numerical vertical profile data. To ensure comparability, both the observational and modeling data are interpolated to a vertical resolution of 10 meters. The models evaluated include: a) GFS with a spatial resolution of 28 km, b) ICON with a spatial resolution of 13 km, and c) AAROME with a spatial resolution of 2.5 km.

---

## Author Comment (AC4)

**Review**

This paper is an ambitious and very timely study that introduces a novel methodology for in-plume radiosonde profiling during wildfires. The paper offers new insights into fire–atmosphere interactions and pyroconvection dynamics, which is an active area of research that needs further exploring. The indicators of potential transitions to extreme behavior help identify plume top heights and characterize pyroconvection prototypes, which are notable contributions that will help fire management better understand fire behavior. The authors also provide one of the most extensive datasets of simultaneous ambient and in-plume profiles to date. While the paper offers key contributions and advancements, the authors must refine the write-up to be more consistent and coherent, improve the visualization of figures, and clarify ambiguous points in the text to enhance readability. Please find comments and suggestions below:

**Clarify/highlight key contributions:**

● I suggest restructuring the early stages of the paper to highlight key contributions and advancements more clearly in the beginning. For example, line 466: "Observing the plume top dilution just below the Lifting Condensation Level (LCL) in real-time is a unique and valuable aspect of this methodology."

*Thanks for the comment.*

*Yes, in the revised version, we have proposed a more comprehensive description of the state of the art. We have placed this methodology in a better context, enabling us to determine, using real-time state variables, whether pyroconvection would occur. As such, these new data and its analysis in conserved variables provide reliable information for informed strategic, tactical, and safety decision-making.*

*In response to the author's comment in line 466, we changed the text:*

> *Old version:*
>
> *Unlike the Rojals fire, this case shows evidence of a potential transition from a convective plume prototype to an overshooting pyroCu prototype, as suggested by MU parcel. Observing the plume top dilution just below the Lifting Condensation Level (LCL) in real-time is a unique and valuable aspect of this methodology. Thanks to these in-situ profiles, crews left for safety zones, and 2 hours later, the formation of an opyroCu worsened the spread of the fire.*
>
> *New version:*
>
> *In contrast to the Rojals fire, this case provides evidence of a potential transition from a convective plume to an overshooting pyroCu prototype, as indicated by the MU parcel. Real-time observations revealed that the plume top was close to the lifting condensation level (LCL), even though the fireline intensity was moderate at that moment. This observation, along with firefighters' reports of an increasing rate of fire spread, alerted us to a possible sudden and dangerous change in fire behavior, catching the firefighters off guard. There, the in-situ profiles and their analysis using a systematic criteria of pyrcoumulus formation (Castellnou et al., 2022) were key to provide warnings and take decisions to movethe crews to safety zones. Two hours later, the formation of a pyroCu confirmed the expected intensification of the fire. This aspect of the methodology is both unique and valuable, as it enables proactive tactical adjustments to enhance safety.*

*Following the comment to address the firefighter safety in the paper, and as proposed in the answer to RC1 comments, we have added a table clarifying how the information has been used for decision-making.*

*Indeed, to highlight the key safety contributions of the proposed methodology in each case shown in the paper, we have added safety/awareness information to the tables accompanying Figures 3 and 9-12. The new information describes how the in-situ radiosondes confirmed or added new information to the analysis that the numerical model data couldn't provide.*

| Fire event | Region | Fuel | in-plume hour (UTC) | Ambient hour (UTC) | area (ha), (Total / hour) | FLI (kW·m-1) | ROS (m·s-1) | FRE (TJ) | Prototype |
|---|---|---|---|---|---|---|---|---|---|
| Granja d'Escarp 03-07-2024 | ME | TU5 | 16:37 | 16:58 | 118 / 36 | 26741 | 1.05 | 2.1 | Convective |
| | *Sonde confirms model proposed no pyroconvection transition.* | | | | | | | | |
| La Selva del Camp 03-08-2023 | ME | SH5 | 15:33 | 15:50 | 3.2 / 0.09 | 1258 | 0.029 | n. d. | Surface |
| | *Sonde identified an unexpected (by numerical model) pyroCu potential, prompting monitoring and safety debriefing for firefighters. This change modified tactical priorities.* | | | | | | | | |

● The title and abstract imply that the paper holds important implications for firefighter safety. This is also foreshadowed in Lines 79-80 but not directly addressed in the paper. Currently, it seems that safety is only discussed in terms of data collection. I recommend including a sub-section or paragraph that discusses how the paper's methods and results can be used for firefighter safety.

*We value this comment. The paper focuses on providing a safe methodology for gathering detailed and in situ information about the pyroconvection prototype and the transition between prototypes. Awareness of such processes is crucial for ensuring the safety of firefighters and civilians in the fire area.*

*The current operational analysis processes, although improved from their previous state, are still based on numerical model data that is uncertain in terms of potential changes of the weather due to local phenomena and/or the wildfire-weather interaction (see answer to Pedro Oria and the new proposed Figure S8) or utilize satellite geostationary information that can only provide reactive safety to ongoing processes.*

*In the text, we have insisted on the use of the proposed methodology in:*

a) *Ambient radiosondes complement the numerical models results by adding almost real time and the local meteorological conditions, so the analysis is more robust since it incorporates the local geography and weather-induced change due to the interaction of fire or local meteorology*
b) *In-plume sondes can register the fire-induced changes in ABL and LCL levels. Therefore, we have a more accurate quantification of changes in height and in the time of the fire-atmosphere coupled situation. This information can be contrasted with available numerical results and improve our predictions in a very short time*
c) *Importantly, the information is ready to be used directly on the fireline by the incident management team*

*In the previous comment, we have reinforced the safety approach by adding the accompanying tables to Figures 3 and 9-12*

*Also, as stated in the previous comment, in the answer to RC1 we include a table in the available data signaling the use of every sonde information for decision making, classifying it for safety, strategy, tactics. Together, the new table and the new information in the figures added focus in sondes use for firefighter safety and highlights how sondes in our campaigns have been used for incident management teams. This new information is summarized in new section 3.4, that is renamed from 'Failed profiles' to 'Usability of plume profiling methodology'*

*In this new text, we also addressed concerns about safety for sonde launching during ongoing pyroCb events and their complex conditions due to downdraft-driven chaotic fire behavior.*

*Old text:*

*'3.4 Failed profiles*

*It is important to note that during the campaigns, we did not observe detrained sondes from the plume once the sonde entered the plume neck. However, we have had cases of sondes failing to enter the plume or entering the plume at higher altitudes when we launch into weak or intermittent indraft conditions. Those cases have always been reported with launching conditions too far away from the head fire (Figure S4) or when we launch into a decaying head fire, and there are strong surface winds present (>6 m·s-1). '*

*New text:*

*'3.4 Usability of Plume Profiling for Incident Management in Extreme Fire Events*

*During the five years of fire campaigns conducted from 2021 to 2025, we compiled and analyzed data that clearly supported our methodology of using paired ambient-in-plume profiling with radiosondes on active wildfires (see Table 4). The methodology demonstrated a low failure rate of 7.7% and was instrumental in adapting operational tactics in 39.7% of our case studies. Additionally, it enabled us to safely retire all firefighters to the safety zone in 7% of the cases.*

*It's important to note that during the campaigns, sondes that failed to enter the plume updraft typically did so because they were launched too far from the plume base, landing in weak or intermittent indrafts (see Figure S4), unable to keep them in the indraft flow to the base of the plume and its updraft. This often occurred in the head or flank indrafts. In contrast, rear indraft sondes, which travel in the main indraft into the plume, are normally able to reach the plume updraft and deliver a full plume profiling.*

*This finding is particularly significant in the context of extreme pyroconvective fires. Deep plumes and fully developed pyroconvective clouds (pyroCbs) create stronger rear indrafts, which facilitate the safe launch of sondes from distances that ensure the safety of the launching team. This phenomenon was observed during rear indraft in-plume launches of sondes in the Santa Coloma Queralt 2021 fire (SCQ21) and the Guissona 2025 fire (Gui25). In both instances, the sondes were launched at 2.2 km (SCQ21) and 9.3 km (Gui25) into a strong rear indraft, successfully entering the plume and reaching altitudes above 8,000 meters.*

*Table 4.- Summary of success and use in decision making of the sondes launched ).*

| Type of sonde | Proportion over total sondes | description | |
|---|---|---|---|
| Failed sondes | 7.73% | 61.3% too weak indraft, or launching too far away | |
| | | 23% pushed to the ground by rear indraft | |
| | | 15.3% due to sonde failure | |
| operational | 73.27% | Awareness | 34.1% |
| | | Tactical | 32.7% |
| | | Safety | 7% |
| Research | 19% | | |

**Improving the visualization of main figures:**

● Figure 1: The proportion of the observations are difficult to differentiate. I suggest adding a text label denoting the relative proportion for each country/region (e.g., ME, 44.73%, AE, 3.29%, SA, 51.98%). Also, since the

symbols are difficult to see, it would help to see the distribution of observations by fire type and by country/region (e.g., additional stacked bar plots)

*Thanks for helping to improve the Figures. Yes, we have added the labels and differentiated fire types. We also added the extra sondes launched during the 2025 European fire season. The new Figure will look:*

[Figure]

**Figure 1**: Location of the 173 in-plume profile observations during the radiosonde campaigns conducted between 2021 and 2025. Sondes are identified based on whether they were launched during wildfires (circle) or prescribed fires (triangle). The color of each dot represents the campaign year, while the size of the dot reflects the total fire size (in hectares). The distribution of the sondes by fire sizes, region, and type of fire (wildfire or prescribed fire) is shown in the bar plots to highlight the range in which the methodology has been tested. Last updated: September 15th, 2025.

● Figure 3: Please update the horizontal axis labels so that they are consistent (i.e., Panel a): θ (K) and Panel e): Temperature (oC))

*Thanks for spotting the detail. We have solved the issue*

[Figure]

● Figure 4: Please update the bottom table of the figure. I suggest to rename the contents of "Fire event", expand "Reg" (i.e., region), and write the fuel model code instead of as a number. Also, under "Ambient Hour", there are two times for Granja d'Escarp.

*We have solved the issue and translated the table changes to Figures 9, 10, 11, and 12. Now the figures are consistent. This is shown in the previous comment about the table in the first comment of this reviewer*

● Figure 8: I suggest updating the legend for plot a). First, I suggest using the full names of the prototypes. Second, I suggest changing the font formatting of the legend and axis labels to be more visible. Please also edit the x-axis label. Third, I suggest re-ordering/modifying the visibility of the symbols (e.g., Re-order so symbols appear on top or change the transparency). Fourth, for plot b), please edit the typo in the legend to "radar echotop"

*We solved the issue on the Figure, adding the changes to the proposed modifications of RC1:*

[Figure]

● Figures 9-12: Please edit the spacing of the plots in a) and b). Currently, the x-axis labels are overlapping. Also, please edit the "Fire event" names,

*We have solved the issue by incrementing the spacing between subplots and tilting the x-axis labels. We provide an example for the Figure 9b profiles. The rest of the plots in Figure 9-12 have been updated accordingly:*

[Figure]

● Figure 12: In the legend, the time formatting is incorrect (Check AM and PM) and the line symbol for "Fire spread 19pm to 2am" is missing.

*We have solved the issue, the right figure is:*

[Figure]

**Clarification**

● In Line 171, why should the soundings be taken "no more than 1 hour apart"? Is there any supporting citation or any reasoning?

*Thanks for the comment, we updated our text.*

***Old version:***

*'Our strategy and primary objective were to systematically obtain (1) an ambient sonde outside the shading of the plume and (2) an in-plume sonde, launched close enough to the plume into the indraft, capturing the fire-induced changes in the atmospheric boundary layer (ABL). Both soundings should be taken no more than 1 hour apart (Figure 2)'.*

***New version:***

*'Our strategy and primary objective are to systematically obtain (1) an ambient sonde outside the influence of the wildfire and (2) an in-plume sonde, launched into the updraft, capturing the fire-induced changes in the atmospheric boundary layer (ABL). Soundings should be taken no more than 1 hour apart (Figure 2) due to the ABL's response time of approximately one hour or less (Granados et al. 2012, Liu & Liang 2010, Stull, 1988). This maximum time ensures that the ambient and in-plume soundings remain comparable'.*

● Line 424: Why was the estimated plume top defined as the maximum height where radar reflectivity was equal or higher than 12 dBz?

*In section 2.5.2 of the methodology, we have clarified the echotop value used to filter the radar data. Our decision was based on the work of Krishna et al. (2024), which suggests that the plume top should be retrieved at an echotop of 10 dBz or higher. To ensure safety, we chose a threshold of 12 dBz.*

*Krishna, M., Saide, P. E., Ye, X., Turney, F. A., Hair, J. W., Fenn, M., & Shingler, T. (2024). Evaluation of wildfire plume injection heights estimated from operational weather radar observations using airborne Lidar retrievals. Journal of Geophysical Research: Atmospheres, 129(9), e2023JD039926.*

***Old version:***

**'2.5.2 Data collection for post-analysis and research**

- *Radar measured echotop. It is a proxy measure for the plume top. We analyze the radar echotop height (m) using radar data from the Servei Català de Meteorologia (www.meteo.cat). We filter the radar echotop data and define the estimated plume top as the maximum height where the reflectivity value equals or is higher than 12 dBZ. Unfortunately, the data for all fires is not available. This dataset is utilized to validate the estimates of plume tops collected from in-plume radiosondes during 18 wildfires'.*

***New version:***

**'2.5.2 Data collection for post-analysis and research**

- *Radar measured echotop. It is a proxy measure for the plume top. We analyze the radar echotop height (m) using radar data from the Servei Català de Meteorologia (www.meteo.cat). The data treatment is the following. We filter the radar echotop data and define the estimated plume top as the maximum height where the reflectivity value equals or is higher than 12 dBZ (Krishna et al, 2023). Unfortunately, the data for all fires is not available. This dataset is utilized to validate the estimates of plume tops collected from in-plume radiosondes during 18 wildfires'.*

● Line 449: It is difficult to recognize the "excess of 7K". I suggest explaining the surface temperature values or labeling them to make this observation clearer

*We have labeled the T$^a$ excess on the plot for clarity (see improved final Figures in previous comment about Figures 3 and 9-12)*

● Line 454: The authors claim that "the theoretical undiluted updraft height, estimated using the MU parcel method (black dashed arrow), is located at 980 m AGL". However, Figure 9A shows that the black dashed arrow lies above 1000 m AGL. There are many lines and colors in the plots, which make the figure difficult to understand. While the figure captions seem well-explained, the authors should clearly define each line and color as well as provide ample reasoning for deciding on specific values (e.g., 980m AGL) in the text to enhance readability.

*We have reorganized the legend and lines in the plots to clarify its reading. It is shown in previous comments about Figures 3 and 9-12*

**Minor Comments**

● Explicitly refer to ICON-EU as the atmospheric model

*In section 2.4, we complement the text to identify ICON_EU as the atmospheric model when possible. It is not available in South America, where we shift to ICON (13*13 km):*

**Old version***:*

- *Framing the day vertical atmospheric profile conditions:*

*We utilize the ICON-EU 7\*7 km2 resolution simulated atmospheric vertical profile to understand the general conditions we can expect (https://www.dwd.de/EN/ourservices/nwp_forecast_data/nwp_forecast_data.html).*

**New version:**

- *Framing the day vertical atmospheric profile conditions with atmospheric numerical models:*

*We utilize the ICON (13 km$^2$ horizontal resolution) global model as a reference. In using the meteorlogical values in Europe with ICON-EU, we use a finer 7 km$^2$ resolution. The modeled atmospheric profile identifies the general conditions, without local topography or fire-induced changes, we can expect in the fire area (https://www.dwd.de/EN/ourservices/nwp_forecast_data/nwp_forecast_data.html).*

*Also, in the text, every time we refer to the ICON we make sure we refer to ICON-EU or ICON accordingly*

● Please add the short-form abbreviations of each prototype in Table 3. I suggest adding them inside brackets below each prototype under the column "Pyroconvective Prototype"

*Done*

● Please explain the state variables recorded by the in-plume (updraft) sonde or explicitly refer to Table 2. The authors state that the in-plume sonde is classified based on its position (head, flank, rear). Does this imply that, ideally, users should launch individual sondes at each position to address turbulence experienced near the head direction of the fire?

*The estate variables are the same for the ambient as the in-plume sondes. Those are explained in point 2.5.1 and complemented in appendix S3.*

*In regard to the in-plume (updraft), we recognize that different launch positions around the plume can capture varying induced indrafts at the surface, as discussed in section 2.4 and illustrated in Figure 2. We differentiate*

*the different launching positions by the type of induced indrafts in the surface that will transport the sonde into the plume updraft, as discussed in section 2.4 and illustrated in Figure 2, to analyze them separately. This will allow us to assess each launching position sensitivity on gathering the information needed for the awareness of the pyroconvection prototypes transition.*

***Old version:***

- *In-plume or updraft sonde:*

*Launched near the flame front into the plumes' indraft, the device measures state variables affected by the fire-atmosphere interaction. However, turbulence around the head of the fire can significantly impact the readings. To address this issue, we classified each updraft sonde based on its position relative to the plume's indraft, using categories: head indraft (downwind from head fire front), flank indraft (on the flanks), or rear indraft (upwind from the head fire front) launching positions (Figure 2). This classification ensures interoperability among sondes of the same kind of indraft.*

***New version:***

- *In-plume (updraft) sonde:*

*Launched near the flame front into the indraft of the plume, the device is pulled into the plume and rises with the updraft. It measures the state variables affected by the turbulent interactions between the fire and the atmosphere. The intensity of the indraft and turbulence varies significantly from the head to the rear and flanks of the fire. To analyze the sondes sensitivity on capturing the characteristics of the plume updraft for the different launching positions, we classified each in-plume sonde by its launch type (Figure 2): head indraft (downwind from the fire front), flank indraft (on the flanks), and rear indraft (upwind). This classification ensures the interoperability of sondes within the same indraft.*

● Please use consistent terminology in the paper. For instance, in Figure 7, I suggest using "Ambient" and "In-plume" (instead of "Environment").

Solved:

[Figure]

● Please provide a citation or reference to "the previous analysis" on Line 420

*Thanks for the observation. We propose changes to clarify the text:*

***Old version:***

*'Based on rising velocities, the previous analysis provided a first-order estimate of the plume's top height'.*

***New version:***

*Our analysis of the rise velocity porofiles as presented in Figures 6 and 7 shows us that this variable is a first-order criterion to estimate the top height of the plume the plume's top height'.*

***New Bibliography due to this revision***

*Granados-Muñoz, M. J., Navas-Guzmán, F., Bravo-Aranda, J. A., Guerrero-Rascado, J. L., Lyamani, H., Fernández-Gálvez, J., & Alados-Arboledas, L. (2012). Automatic determination of the planetary boundary layer height using lidar: One-year analysis over southeastern Spain. Journal of Geophysical Research: Atmospheres, 117(D18).*

*Krishna, M., Saide, P. E., Ye, X., Turney, F. A., Hair, J. W., Fenn, M., & Shingler, T. (2024). Evaluation of wildfire plume injection heights estimated from operational weather radar observations using airborne Lidar retrievals. Journal of Geophysical Research: Atmospheres, 129(9), e2023JD039926.*

---

## Author Comment (AC5)

The paper presents an operational and research-oriented observation method for in-plume radiosonde profiling during extreme wildfire events. It combines direct fireline observations with atmospheric soundings to quantify fire-atmosphere effects and evaluate pyroconvective transitions in real time. The authors should be commended for their long-term field effort—150 sondes over multiple fire seasons and continents—and for demonstrating the feasibility of affordable, lightweight instrumentation for operational plume monitoring. The work addresses a long-standing gap between model-based indices of pyroconvection and field observations available to incident managers. Congratulations for the work, it is obviously a very valuable field work analyzed here, there are no other consistent direct observations dataset of so many plumes to my knowledge.

The paper is generally well structured, clearly written, and sound. It provides valuable insight into how in-plume thermodynamic profiles can be used to characterize the Atmospheric Boundary Layer (ABL), plume dilution, and potential transitions from dry to moist pyroconvection. The dataset has high potential value for model validation (e.g., Micro-HH, Meso-NH/ForeFire, WRF-Sfire) and for improving fire awareness protocols in operations. The figures are instructive, and the field documentation is impressive. The manuscript will interest both fire scientists and operational meteorologists.

> *We appreciate the reviewer's positive and encouraging feedback. We are pleased that the reviewer recognizes the significance of our long-term field effort and the potential of our dataset for both research and operational applications. Our primary goal has been to connect model-based indices with real-time field observations, and we are glad that this contribution is valued by both communities.*
>
> *We will address all specific comments and suggestions raised by the reviewer in the following point-by-point responses.*
>
> *We will update the data availability files to ensure this aspect is well explained.*
>
> *Old version:*
>
> ### *Data availability*
>
> *Final Dataset in EWED project data portal: http://wildfiredataportal.eu/*
>
> *The profiles in the Figures are in DOI 10.5281/zenodo.15264835*
>
> *New version:*
>
> ### *Data availability*
>
> *To facilitate the use in research of the in-plume radiosonde data, the dataset is organized in a data portal that includes (1) radiosonde file observations, (2) fire-spread isochrones, (3) perimeters for each fire, and(4) field-captured plume images of plumes analyzed. The information is georeferenced to facilitate further analysis with reanalysis datasets.*

*Final Dataset in EWED project data portal:* [http://wildfiredataportal.eu/](http://wildfiredataportal.eu/)*. Please, note It is still not operational until December 2025*

*In addition to the live data portal, the paper used radiosonde files are in DOI 10.5281/zenodo.15264835*

Some minor comments: is there any way to perform quantitative uncertainty associated with in-plume sondes ? representativeness in turbulent regimes and the sensitivity of plume-top estimates, maybe discuss that (a radiosonde is a single point in space / time).

*Thanks for pointing out the need for this analysis to complement our research. Following similar comments made by CC1, we have performed a simple yet insightful uncertainty analysis by comparing sondes launched simultaneously.*

*In short, for the sondes that observed the state variables of the ambient around the fire, we analyze 5 sondes launched simultaneously (Figure S7.1).*

*For in-plume sondes, we have conducted the same analysis for those fires where we had simultaneous sondes launched (within 30 minutes of each other). We show the Casablanca III Chilean fire case in Figure S7.2*

*We have updated the proposed S7 complementary material in response to the CC1 comments by adding an uncertainty analysis of the radiosonde-plume top derived from simultaneously launched sondes. Briefly, we have normalized the sondes by height, potential temperature, and relative humidity. We have compared the mean and standard deviation of the aggregated dataset in Figure S7.3.*

***This new section has been included in the supplementary material:***

***S7.-Uncertainty assessment for the radiosonding system***

*To quantify the uncertainty in our observations from the sounding due to different trajectories, we calculated the mean and standard deviation along the vertical profile for each variable based on simultaneous sondes launched at the same location.*

*1.- Uncertainty in vertical profile measured variables*

*As shown in Table S7.1, the uncertainty observed is reduced below 1K in Ɵ, 2% in RH, and 2 m·s$^{-1}$ in wind speed. The maximum uncertainty level is 3.64 K in Ɵ, 7.19 in RH, and 2.43 m·s$^{-1}$ in vertical velocity. This maximum uncertainty is primarily located at the top of the mixed layer (grey shadow in Figures S7.1 and S7.2), identified as ABL top for the ambient conditions in Figure S7.1 and plume top for the in-plume conditions in Figure S7.2. This level of uncertainty is typical, as both the ABL and plume top are influenced by turbulent motions and, therefore, influenced by fluctuations.*

***Table S7.1****. Uncertainty analysis of simultaneous radiosonde trajectory for ambient and in-plume measures for the variables used in the radiosounding methodology: Ɵv (K) as virtual potential temperature, RH (%) as relative humidity, WS (m·s$^{-1}$) as wind speed, WD (º) as wind direction and in the case of in-plume sondes vertical velocity (m·s$^{-1}$).*

| Type | fire | | σ Θv (K) | σ RH (%) | σ WS (m·s⁻¹) | σ WD (º) | σ Vertical velocity (m·s⁻¹) |
|------|------|------|---------|---------|---------|---------|---------|
| Ambient | Tivissa 08-08-2025 | mean | 0.39 | 1.61 | 1.13 | 12.89 | |
| | | max | 2.64 | 13.20 | 2.42 | 21.22 | |
| In-plume | Casablanca III 08-02-2023 | mean | 0.78 | 1.12 | 1.18 | 34.85 | 0.84 |
| | | max | 2.57 | 7.29 | 2.51 | 143.92 | 3.26 |
| In-plume | Granyena 987 ha 21-06-2025 | mean | 0.41 | 3.48 | 1.60 | | 0.53 |
| | | max | 1.17 | 13.5 | 3.10 | | 1.31 |
| In-plume | Pauls 3800 ha 07-07-2025 | mean | 0.93 | 2.53 | 2.04 | 13.47 | 1.08 |
| | | max | 6.33 | 9.54 | 5.81 | 132.63 | 5.49 |
| In-plume | Casablanca III 12073 ha 10-02-2023 | mean | 0.52 | 1.30 | 1.19 | 63 | 0.32 |
| | | max | 5.90 | 6.16 | 2.48 | 136 | 1.77 |
| In-plume | Casablanca III 12073 ha 10-02-2023 | mean | 0.95 | 1.49 | 1.63 | 22.94 | 0.88 |
| | | max | 4.18 | 6.17 | 2.51 | 61.56 | 2.31 |
| In-plume | Tortosa 280 ha 28-05-2024 | mean | 0.58 | 3.43 | 1.57 | 23.47 | 0.73 |
| | | max | 5.59 | 9.97 | 2.46 | 60.18 | 2.37 |
| In-plume | Manuel Rodriguez 370 ha 05-02-2025 | mean | 0.33 | 0.47 | 1.13 | 11.12 | 0.45 |
| | | max | 3.16 | 7.95 | 2.73 | 15.87 | 1.77 |
| In-plume | Patagual 218 ha 08-02-2025 | mean | 0.27 | 1.27 | 1.12 | 10.93 | 0.28 |
| | | max | 2.88 | 6.81 | 2.42 | 21.75 | 1.14 |
| In-plume | Vega Honda 773 ha 09-02-2025 | mean | 1.17 | 1.39 | 1.02 | 13.11 | 0.69 |
| | | max | 4.80 | 2.67 | 2.60 | 22.95 | 2.43 |
| | Aggregated mean | | 0.59 | 1.55 | 1.35 | 19.97 | 0.58 |
| | Aggregated max | | 3.64 | 7.19 | 2.83 | 54.35 | 2.18 |

[Figure]

**Figure S7.1**.-Uncertainty analysis is indicated by the bars surrounding the mean (dot) values of the profile observations taken under ambient conditions: virtual potential temperature (Θv, K), relative

humidity (RH, %), wind speed (WS, m·s$^{-1}$), and wind direction (WD, degrees). This analysis involves calculating the uncertainty in the vertical profile measurements based on five different radiosonde trajectories launched from the same location between 16:03 and 16:11 UTC. The grey shadow area represents the uncertainty in the height estimation of ABL top.

[Figure]

**Figure S7.2.**-*Uncertainty analysis is indicated by the bars surrounding the mean (dot) values of the in-plume profile observations: virtual potential temperature (Өv, K), relative humidity (RH, %), wind speed (WS, m·s$^{-1}$), wind direction (WD, degrees) and vertical velocity (w, m·s$^{-1}$). This analysis involves calculating the uncertainty in vertical profile measurements from 3 radiosondes launched at the same location between 21:46 and 21:51 UTC during the Casablanca fire (Chile) (see Figure 11). The grey shadow area represents uncertainty in the height estimation of plume top.*

**2.- Uncertainty in plume top height**

Fluctuations or uncertainties in the plume top height can produce different plume top estimations. Those fluctuations have been quantified as absolute and relative error in the sondes launched simultaneously (Table S7.2) resulting in an aggregated mean absolute error of 114.4 m, with a maximum of 282 m and a standard deviation of 81.6 m.

**Table S7.2**. *Uncertainty analysis of plume top assessment by radiosonde trajectory for in-plume measures. Based on vertical velocity estimation of plume top for every sonde, we obtain the average plume top, the standard deviation, the absolute error or difference, and the relative error.*

| Fire | Date | Sonda 1 plume top (m) | Sonda 2 plume top (m) | Average (m) | Standard deviation (m) | Absolute error (m) | Relative error |
|------|------|------|------|------|------|------|------|
| Granyena | 21-06-2025 | 2744 | 2583 | 2663,5 | 113,84 | 161 | 0.06 |
| Casablanca III | 08-02-2023 | 1932 | 2015 | 1973,5 | 58,68 | 83 | 0.04 |

| | | | | | | | |
|---|---|---|---|---|---|---|---|
| Pauls | 07-07-2025 | 2633 | 2768 | 2700,5 | 95,45 | 135 | 0.04 |
| Casablanca III | 10-02-2023 | 612 | 894 | 753 | 199,40 | 282 | 0.37 |
| Casablanca III | 10-02-2023 | 1308 | 1378 | 1343 | 49,49 | 70 | 0.05 |
| Tortosa | 28-05-2024 | 1792 | 1751 | 1771,5 | 28,99 | 41 | 0.023 |
| Manuel Rodriguez | 05-02-2025 | 1131 | 1054 | 1092,5 | 54,44 | 77 | 0.07 |
| Patagual | 08-02-2025 | 1348 | 1433 | 1390,5 | 60,10 | 85 | 0.06 |
| Vega Honda | 09-02-2025 | 529 | 634 | 581,5 | 74,24 | 105 | 0.18 |
| | | | | aggregated | **81.6** | **114.4** | **0.1** |

*To better quantify the uncertainty in determining the plume top by sonde trajectories we have computed an agreggated plume top probabilitydistribution (Figure S7.3 ). To aggregate all the different vertical profiles, , we use a normalized vertical profile height that extends twice the height of the measured mixed layer. We also normalized the potential temperature, and relative humidity by each profile mixing layer average.  Using bins of 10% of the normalized height, we compare, for the in-plume sondes in table S7.2, the aggregated mean and standard deviation distribution of  ($\Theta v$ (K), RH (%), and vertical velocity ($m·s^{-1}$).  The obtained probability distribution (Figure S7.3) aligns with the results shown in Figure S7.1 S7.2. It shows that despite single sonde trajectory inside a turbulent plume, the aggregated probability distribution identifies the plume top probability exactly at 100-110% of the normalized height where uncertainty of RH and $\Theta v$ increases inversely to that of vertical velocity . It reliably identifies plume top height. This consistency holds true despite the singular nature of the sonde trajectory and varying fire conditions.*

[Figure]

*Figure S7.3.-- Probability of plume top distribution based on uncertainty in simultaneous in-plume profiles of virtual potential temperature ($\Theta v$, K), relative humidity (RH, %), and vertical velocity (w, $m·s^{-1}$). We analyzed 18 radiosondes across 9 sets of simultaneous launches (within 30 minutes). RH and $\Theta v$ were normalized by the average mixing-layer value, and profile height by the mixing layer height. Uncertainty is quantified as the standard deviation of the mean at every 10% ofthe normalized height. Results show a consistent plume assessment (high plume top probability) between 100-110% of the*

*vertical profile (indicated by the dark blue dashed line), demonstrating the methodology's accuracy despite radiosonde measurement uncertainties.*

Also, among 150+ events, is there any availability on some on weather radar data ? a brief comparison of measured plume heights with radar would help to contextualize accuracy beyond the few examples shown.

> *Unfortunately, as stated in the paper, mobile radar, or a permanent network of radar Doppler, was not available in the regions where we have launched the sondes. However, a new set of radars is being installed in our region that will facilitate such data availability. Indeed, in 2026, we will start deploying mobile radar to the wildfires. These measurements will complement our sounding analysis and will reinforce each other!*

> *In the current study, we used the echotop archive of weather radars to compare the sonde-estimated plume top with 12 dBZ radar echotops (Figure 8a). The figure enables us to quantify the uncertainty of plume top estimation when comparing radar data with radiosonde to an average error of 166.82 m. However, it is important to note that the divergence between radiosonde and radar-measured heights increases with plume top height above 6000 m AGL.*

> *In the revised manuscript, and following the comments of CC5, we have clarified the issue and cited the previous work of reference:*

> *Krishna, M., Saide, P. E., Ye, X., Turney, F. A., Hair, J. W., Fenn, M., & Shingler, T. (2024). Evaluation of wildfire plume injection heights estimated from operational weather radar observations using airborne Lidar retrievals. Journal of Geophysical Research: Atmospheres, 129(9), e2023JD039926.*

> ### *Old version:*

> ### *'2.5.2 Data collection for post-analysis and research*

> - *Radar measured echotop. It is a proxy measure for the plume top. We analyze the radar echotop height (m) using radar data from the Servei Català de Meteorologia (www.meteo.cat). We filter the radar echotop data and define the estimated plume top as the maximum height where the reflectivity value equals or is higher than 12 dBZ. Unfortunately, the data for all fires is not available. This dataset is utilized to validate the estimates of plume tops collected from in-plume radiosondes during 18 wildfires'.*

> ### *New version:*

> ### *'2.5.2 Data collection for post-analysis and research*

> - *Radar measured echotop. It is a proxy measure for the plume top height. We analyze the radar echotop height (m) using radar data from the Servei Català de Meteorologia (www.meteo.cat). We filter the radar echotop data and define the estimated plume top as the maximum height at which the reflectivity equals or exceeds 12 dBZ (Krishna et al., 2023). Unfortunately, the data for all fires is not available. However, the available dataset is utilized to validate and to corroborate*

*the estimates of plume tops collected from in-plume radiosondes during 18 wildfires.*

Minor:

Ensure consistent notation for potential temperature (θ) and virtual potential temperature (θv).

line 467 - "opyroCu" with "pyroCu"

*Thanks for the comment. We have gone through the document again and ensured consistency on such terms.*

This is a well-executed and highly relevant contribution that bridges operational practice and research in pyroconvection monitoring. I recommend minor revision to address the small editorial and methodological clarifications listed above. Once revised, the manuscript will constitute an important reference for both field operations and coupled fire-atmosphere modeling.

---

## Author Response (AR2)

The authors should make the following (technical) corrections, after which the paper can be published:

*We answer the editor comments point by point*

L24: coupling fire-atmospheric --> fire-atmospheric coupling

*Done*

L39: there should be a space between problem and (Cardil...

*Done*

L88: have been used --> has been used

*Done*

L169: ( video --> (video

*Done*

L216 and elsewhere: consider to write thetav as theta_v (subscript v)

*Done*

L230: subsection heading should be in boldface

*Done*

L253 and elsewhere: should Td and Ts not be written with subscripts?

*Done*

L263: RH. (Li ... --> RH (Li ...

*Done*

L267: height(Castellnou --> height (Castellnou

*Done*

L357: increase ...by 3 K is typed in purple, should be black

*Done*

L441: Figures S8.1 to S8.3 should not be highlighted text

*Done*

L445: using is also typed in purple

*Done*

L630: withastanded --> withstood

*Done*

L635: indraftinto --> indraft into

*Done*

L675: cores , --> cores,

*Done*

L723: level. (Figure 12). --> level (Figure 12).

*Done*

Figure 4: make clear in the figure (via the legend) which colours or linestyles indicate Td and Ts from the ICON model and which different colours/linestyles indicate Td and Ts from the sonde observations. There is ambiguity now in interpreting the figure.

*We made the figure again, adding a clear legend in each plot*

L33: clarify what "spotting" means

*We clarify it by changing the text:*

*Old version:*

*Pyroconvection is a key driver in the escalation from wildfires to extreme wildfire events. While dry convection plumes effectively accelerate fire spread and produce long distance spotting , it is the development of moist pyroconvection plumes by the formation of pyrocumulus and pyrocumulonimbus (pyroCu/Cb, AMS, 2023) that dramatically intensifies fire behavior. Deep pyroCu/Cb events amplify dry pyroconvective plume dynamics through powerful indrafts and downdrafts, triggering chaotic surges in spread rate, increasing massive and long-range spotting on the head and flanks, and generating deep flames and vortices (McRae et al., 2015; Peterson et al., 2017)*

*New versión:*

*Pyroconvection is a key driver in the escalation from wildfires to extreme wildfire events. While dry convection plumes effectively accelerate fire spread , it is the development of moist pyroconvection plumes by the formation of pyrocumulus and pyrocumulonimbus (pyroCu/Cb, AMS, 2023) that dramatically intensifies fire behavior. Deep pyroCu/Cb events amplify dry pyroconvective plume dynamics through powerful indrafts and downdrafts, triggering chaotic surges in spread rate, increasing massive and long-range spotting (embers ignite new fires at a distance) on the head and flanks, and generating deep flames and vortices (McRae et al., 2015; Peterson et al., 2017).*

*We added the dataset in a repository: 10.5281/zenodo.17886250. Such a repository contains the in-plume and the ambient sondes following the ID in Table S1*

*We solved the supplementary materials formatting issues and modified the main text accordingly to reference the supplementary Figures correctly*